# Advancing image segmentation with DBO-Otsu: Addressing rubber tree diseases through enhanced threshold techniques

**Zhenjing Xie**[ID][☯], **Jinran Wu**[☯], **Weirui Tang**, **Yongna Liu**[ID]*

Tropical Agriculture and Forestry College, Hainan University, Haikou, Hainan Province, China

☯ These authors contributed equally to this work.
* 1093061239@qq.com

**Data Availability Statement:** All relevant data are within the paper and its Supporting information files.

## Abstract

Addressing the profound impact of Tapping Panel Dryness (TPD) on yield and quality in the global rubber industry, this study introduces a cutting-edge Otsu threshold segmentation technique, enhanced by Dung Beetle Optimization (DBO-Otsu). This innovative approach optimizes the segmentation threshold combination by accelerating convergence and diversifying search methodologies. Following initial segmentation, TPD severity levels are meticulously assessed using morphological characteristics, enabling precise determination of optimal thresholds for final segmentation. The efficacy of DBO-Otsu is rigorously evaluated against mainstream benchmarks like Peak Signal-to-Noise Ratio (PSNR), Structural Similarity Index (SSIM), and Feature Similarity Index (FSIM), and compared with six contemporary swarm intelligence algorithms. The findings reveal that DBO-Otsu substantially surpasses its counterparts in image segmentation quality and processing speed. Further empirical analysis on a dataset comprising TPD cases from level 1 to 5 underscores the algorithm's practical utility, achieving an impressive 80% accuracy in severity level identification and underscoring its potential for TPD image segmentation and recognition tasks.

## 1 Introduction

### 1.1 Rubber tree tapping panel dryness

The rubber tree, a pivotal economic crop, significantly contributes to the global economy with its primary product, natural rubber [1]. Growth, latex yield, and health of rubber trees are vulnerable to various environmental and human-induced factors [2–4]. Tapping Panel Dryness (TPD) poses a major challenge during latex extraction, leading to latex tube degeneration and considerable yield reduction [3, 5–8]. Unraveling the mechanisms underlying TPD is essential for boosting resilience and latex production in rubber trees and for the sustainable evolution of the rubber industry [9, 10]. Recent advancements in the molecular understanding of TPD [2, 7, 9, 11, 12] have been significant, yet their practical application in disease identification remains hindered by traditional methods. The industry's current reliance on visual inspections by experienced professionals is limited by subjectivity and varying efficiency and accuracy

**Funding:** This research was supported by the Hainan Provincial Natural Science Foundation of China, grant number 20163140. This grant primarily funded the construction of the dataset and the research into the segmentation algorithm.

**Competing interests:** The authors have declared that no competing interests exist.

levels. With extensive global rubber plantations and diverse disease manifestations, traditional manual methods fall short. Image processing technologies emerge as a superior alternative, offering objectivity and robust big data handling, thus providing precise disease assessments essential for research and developing effective disease management strategies in rubber trees.

The fundamental goal in image recognition, especially considering the unique image characteristics of the cuts and latex, is to achieve precise segmentation. High-quality image segmentation directly contributes to enhanced diagnostic accuracy, promoting objectivity and uniformity in evaluations. This approach facilitates automatic disease identification in TPD-affected trees across different stages, assisting researchers in real-time monitoring and future latex yield prediction, and propelling forward the study of rubber tree diseases.

## 1.2 Image segmentation

Image segmentation, an integral component in image processing, lays the technical groundwork for condition diagnosis by isolating image regions with varying characteristics [13]. While traditional segmentation methods primarily leverage threshold setting, histogram analysis [14], region growing, fuzzy clustering [15, 16], K-means clustering [17], and edge detection [18, 19], advanced techniques incorporate active contours, graph cuts, and sophisticated mathematical and probabilistic models [20]. Notably, deep learning approaches [21–25] like Fully Convolutional Networks (FCN) [26],U-Net [27],PSPNet [28] and FC-DenseNet have revolutionized segmentation with their high precision in pixel-level classification. Deep learning methods offer unmatched segmentation accuracy and efficiency; however, automated threshold segmentation techniques remain popular for their simplicity and effectiveness [29, 30]. For example, the multi-threshold Tsallis entropy recursive algorithm by Wang et al. [31] accelerates segmentation while ensuring efficiency. Sharma et al. [32] introduced an optimized multi-level threshold segmentation algorithm, proving its efficacy in brain tumor segmentation and advancing threshold segmentation research. Lei et al. [33] proposed an adaptive granularity Renyi rough entropy method, which augments threshold segmentation accuracy and speed, demonstrating its utility in rapid and efficient image segmentation.

In the context of TPD in rubber plantations, the environmental complexity and signal instability demand more timely and robust recognition technologies. Despite deep learning's superior performance in image segmentation, its high hardware requisites restrict application in wearable devices. Threshold-based segmentation, known for its lightweight and efficient nature, becomes a fitting alternative, especially suitable for wearable device integration, offering vital support for intelligent TPD recognition. Therefore, advancing research and development of these algorithms is imperative for managing rubber tree diseases and facilitating early diagnosis.

## 1.3 Application of Otsu algorithm in image segmentation

The Otsu algorithm, serving as an adaptive threshold segmentation method, has proven highly effective in images with bimodal histograms [34]. Despite its widespread adoption, the algorithm encounters performance limitations in scenarios with highly variable background and target intensity, or significant noise disturbances. For instance, its robustness is compromised in high-noise images affected by salt-and-pepper noise, leading to segmentation inaccuracies [35]. To address these challenges, novel improvements have been proposed. Notably, the integration of the 3D Otsu algorithm with local contrast enhancement has significantly ameliorated segmentation quality while preserving edge details [36]. Methods combining pixel intensity with spatial context, through energy curve optimization, have shown promise under varying lighting conditions, yet they grapple with dynamic environments [37]. Hybrid

algorithms, like the amalgamation of Otsu with K-means clustering, have enhanced accuracy in multi-light spot center detection but impose greater computational demands [38]. Additionally, the 2D Otsu algorithm, when coupled with adaptive energy segmentation and genetic algorithms, demonstrates efficiency, albeit with lingering challenges in handling complex textures and color variations [39, 40].

In this research domain, the application of metaheuristic optimization algorithms is crucial. These algorithms, inspired by natural phenomena and artificial intelligence, are adept at tackling diverse and intricate optimization challenges [41, 42]. Algorithms such as genetic algorithms, Whale Optimization Algorithm (WOA), Particle Swarm Optimization (PSO), and Harris Hawks Optimization (HHO) have each contributed uniquely to threshold selection, each with distinct strengths and weaknesses [43–54]. Recent advancements include swarm intelligence algorithms for multi-threshold segmentation, particularly effective in processing COVID-19 chest X-rays and CT scans [55–59]. Chen et al. [59] augmented the Artificial Bee Colony algorithm with dynamic strategies, boosting initial convergence and global search efficiency. Abualigah et al. [56] innovated with a multi-threshold method based on the Arithmetic Optimization Algorithm (AOA), DAOA, enhancing local search capabilities through differential evolution techniques. Liu et al. [55] merged ant colony optimization with Cauchy mutation and Levy flight strategies, significantly elevating search efficiency and segmentation precision. Emam et al. [60] devised an enhanced Reptile Search Algorithm (mRSA) optimizing both global optimization and image segmentation, showcasing remarkable performance in MRI brain image multi-threshold segmentation. Chen et al. [61] introduced the HVSFLA algorithm, ensuring diverse and active search mechanisms, excelling in multi-threshold segmentation applications for invasive ductal carcinoma of the breast. Abdel-Basset et al. [62] proposed an improved balance optimization algorithm for optimal threshold discovery in grayscale images. These advancements not only propel swarm intelligence applications in medical image processing but also offer potent tools for medical decision-making.

On the other hand, the integration of metaheuristic algorithms with the Otsu method has significantly advanced its capabilities. A study by [63] introduced the DE-GWO-Otsu algorithm, a hybrid of Differential Evolution (DE), Grey Wolf Optimization (GWO), and Otsu's method. This approach addressed the stability and local optima challenges of the GWO. In another innovation, [64] proposed the FOA-Otsu method, merging the Fruit Fly Optimization Algorithm with the Otsu technique, which considerably enhanced real-time image segmentation performance while halving segmentation time. Additionally, [65] developed an Improved Golden Jackal Optimization algorithm (IGJO) integrated with the Otsu method, markedly boosting the accuracy and efficiency in skin cancer image segmentation. The use of the AOA by [66] for determining optimal thresholds in multi-layer segmentation demonstrated effectiveness when coupled with the Otsu fitness function. Furthermore, Rather et al. [67] employed a Levy flight and chaos theory-based Gravity Search Algorithm (LCGSA) to optimize computational efficiency in multi-threshold segmentation, overcoming traditional segmentation issues like local minima and premature convergence. Liu et al. [68] innovated with the HCROA, a primate-inspired WOA, combined with the Chimp Optimization Algorithm, to enhance exploration and exploitation balance, thereby improving segmentation accuracy and noise robustness. Finally, [69] merged Enhanced Fuzzy Elephant Herd Optimization (EFEHO) with the Otsu method, facilitating rapid diagnosis in Alzheimer's disease and Mild Cognitive Impairment (MCI) contexts.

Despite the significant progress made by metaheuristic algorithm-enhanced Otsu methods in various application domains, their robustness [70] and segmentation accuracy remain inadequate when dealing with images containing complex lighting, angles, and texture variations, such as rubber tree tapping scars and latex images. The computational complexity is also

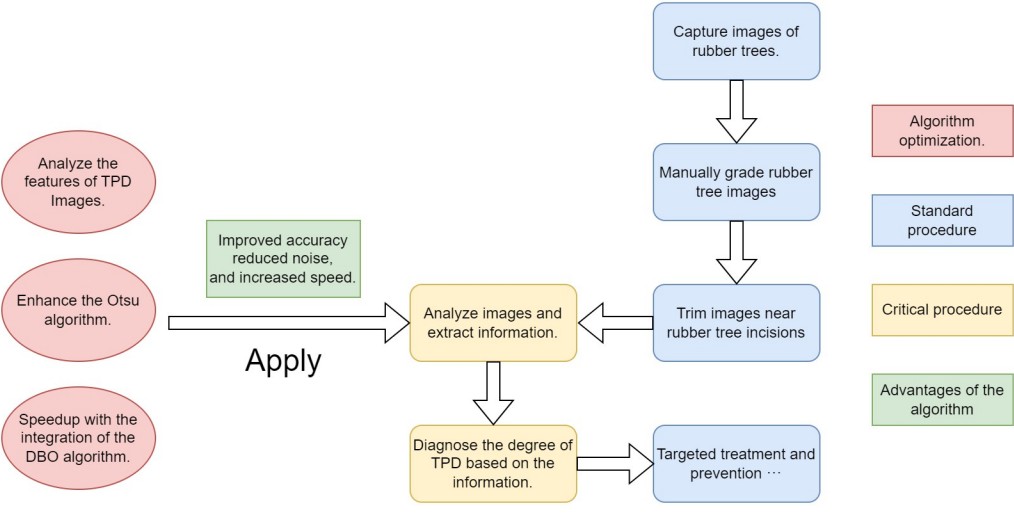

**Fig 1. The improvement and application.**

relatively high. To address this challenge, this study introduces the DBO-Otsu algorithm, a novel integration of the classic Otsu's method with the innovative Dung Beetle Optimizer (DBO), specifically targeting complex image segmentation tasks. The DBO algorithm, inspired by the natural behavior of dung beetles, such as their unique rolling and foraging strategies, effectively enhances search diversity, efficiency, and global convergence [71, 72]. Compared to other metaheuristic algorithms, DBO exhibits pronounced advantages in multi-threshold image segmentation tasks, particularly in terms of convergence speed and solution precision, which are crucial for accurately segmenting TPD in this study. Moreover, the effectiveness of this algorithm has been proven in various practical applications: DBO has demonstrated significant performance improvements in spaceborne SAR image waterbody detection [73, 74], lung cancer detection and classification [75], and pesticide residue identification in rapeseed oil [76]. These successful cases further validate our choice of DBO as the framework for improvement. Additionally, according to the No Free Lunch (NFL) theorem [77], no optimization algorithm excels in all problem types, thus spurring the development of new algorithms and the enhancement of existing ones. Therefore, selecting DBO as the optimization algorithm for this study is based on its unique strengths in addressing specific categories of problems. The advantages of the algorithm are illustrated in Fig 1.

This paper's primary contributions are as follows: (1) The development of the DBO-Otsu algorithm, tailored specifically for complex image segmentation challenges, markedly improving processing efficiency and accuracy. (2) An innovative enhancement of the traditional Otsu method within the DBO-Otsu framework, involving an initial preprocessing stage for multi-threshold segmentation to remove low gray-scale areas, thereby focusing on high gray-scale segments, particularly latex and scars. (3) A comprehensive evaluation of the DBO-Otsu algorithm through a suite of established performance metrics, showcasing its superior performance across various dimensions. (4) An in-depth exploration of the DBO algorithm's application potential in image segmentation, substantiated by practical use cases, notably in diagnosing rubber tree TPD. The paper is organized into subsequent sections as follows: Section 2 elucidates the principles of the Otsu algorithm and the workings of the DBO mechanism; Section 3 elaborates on the DBO-Otsu algorithm's implementation and its innovative aspects relative to the conventional Otsu method; Section 4 demonstrates the algorithm's effectiveness and

comparative analysis via experimental results; Section 5 concludes with a summary of the findings and a discussion on prospective applications.

## 2 Theoretical foundations

### 2.1 Otsu algorithm

The Otsu algorithm is a technique for image binarization segmentation based on global adaptive thresholding [78]. Its core idea revolves around selecting the optimal threshold by calculating the maximum inter-class variance using the gray level histogram of the image [79] Let's consider a digital image of size $M \times N$, containing $L$ distinct gray levels, represented as the set $\{0, 1, 2, \ldots, L - 1\}$. If $n_i$ denotes the number of pixels at gray level $i$, then the image's total pixel count is represented as $MN = n_0 + n_1 + n_2 + \ldots + n_{L-1}$. Consequently, the normalized histogram is defined by the ratio of the pixel count for each gray level to the total pixel count, $p_i = n_i/MN$, from which we have

$$\sum_{i=0}^{L-1} p_i = 1, p_i \geq 0 \tag{1}$$

Consider the threshold $T(k) = k$ where $0 < k < L - 1$. The input image is categorized into two classes: $C_1$ and $C_2$. $C_1$ encompasses pixels with gray values in the range $[0, k]$ while $C_2$ includes those in the range $[k + 1, L - 1]$. Based on this, the probabilities $P_1(k)$ and $P_2(k)$ represent classifications into $C_1$ and $C_2$ respectively.

$$P(k) = \begin{cases} P_1(k) = \displaystyle\sum_{j=0}^{k} p_j \\[2mm] P_2(k) = \displaystyle\sum_{j=k+1}^{L-1} p_j = 1 - P_1(k) \end{cases} \tag{2}$$

For a given threshold value, $T(k)$, we denote the average gray value of pixels in class $C_1$ as $m_1(k)$ and in class $C_2$ as $m_2(k)$. Respectively:

$$\begin{cases} m_1(k) = \dfrac{1}{P_1(k)} \displaystyle\sum_{i=0}^{k} i p_i \\[3mm] m_2(k) = \dfrac{1}{P_2(k)} \displaystyle\sum_{i=k+1}^{L-1} i p_i \end{cases} \tag{3}$$

The average gray level of the image is defined as:

$$m_G = \sum_{i=0}^{L-1} i p_i \tag{4}$$

$\sigma_B^2$ represents the between-class variance, with its formula being:

$$\sigma_B^2 = P_1(m_1 - m_G)^2 + P_2(m_2 - m_G)^2 \tag{5}$$

Again citing $k$, the end result is:

$$\sigma_B^2(k) = \frac{[m_G P_1(k) - m(k)]^2}{P_1(k)[1 - P_1(k)]} \tag{6}$$

Thus the optimal threhold is $k^*$, which maximizes $\sigma_B^2(k)$:

$$\sigma_B^2(k^*) = \max_{0 \leq k \leq L-1} \sigma_B^2(k) \tag{7}$$

For multilevel threshold segmentation, the assumption is that $m$ threshold levels ($t_1$, $t_2$, . . ., $t_m$) segment the image into $m + 1$ categories: $C_0$, $C_1$, $C_2$, . . ., $C_m$. The objective function for the segmentation process is:

$$
\begin{aligned}
J(t)_{max} &= \quad \sigma_0 + \sigma_1 + \cdots + \sigma_m \\
\sigma_0 &= \quad \omega_0 (m_0 - m_G)^2 \\
\sigma_1 &= \quad \omega_1 (m_1 - m_G)^2 \\
&\quad \cdots \\
\sigma_m &= \quad \omega_m (m_m - m_G)^2
\end{aligned}
\tag{8}
$$

From the analysis of the outlined objective function, it becomes clear that the algorithm's solution space extends over a $q - 1$ dimensional realm, with $q$ indicating the total count of thresholds. Within this multidimensional space, specific calculations are crucial, primarily those centered around the inter-class variance, which include determining averages among different classes. Considering these computations, the time complexity for executing multi-level threshold segmentation as per the Otsu method escalates to $O(L^q)$, where $L$ signifies the quantity of gray scale levels. The exhaustive nature of computations across the $q - 1$ dimensional space results in an exponential surge in time complexity relative to the increase in threshold numbers. Contrasting with single-level threshold techniques, multi-level threshold approaches adopt a greater number of thresholds, thereby capturing a more detailed essence of the image. Consequently, while multi-level threshold segmentation furnishes enhanced image detail, it simultaneously amplifies computational complexity. Striking an optimal balance between computational time and segmentation accuracy is imperative, thereby mandating the selection of an apt number of thresholds for effective image segmentation.

## 2.2 DBO algorithm

The position update of the beetle during its rolling behavior can be characterized using a specific mathematical model:

$$
\begin{aligned}
x_i(t + 1) &= x_i(t) + \alpha \times k \times x_i(t - 1) + b \times \Delta x \\
\Delta x &= |x_i(t) - X^\omega|
\end{aligned}
\tag{9}
$$

Let $t$ represent the current iteration number, serving to control the algorithm's iterative process. The symbol $x_i(t)$ denotes the position of the $i$th dung beetle at the $t$th iteration, signifying a candidate solution in the solution space. Additionally, $k \in (0, 0.2]$ represents a deflection coefficient constant, essential for controlling the dung beetle's deflection degree during its search. Another constant, denoted by $b$, belongs to the range (0, 1), where $\alpha$ is a specific

coefficient with values of either -1 or 1 (refer to Algorithm 1). Lastly, $X^\omega$ signifies the global worst position, whereas $\Delta x$ models the changes in light intensity.

**Algorithm 1** Selection strategy for $a$

```
Input:
probability value l
Output:
natural coefficients a
h ← rand(1)
if h > l then
  a ← 1
else
  a ← -1
end if
```

When a dung beetle encounters an obstacle during its rolling phase and is hindered from proceeding, it resorts to a reorientation dance to identify a new direction. Consequently, the position during this dancing behavior is defined by:

$$x_i(t + 1) = x_i(t) + \tan(\theta)|x_i(t) - x_i(t - 1)| \tag{10}$$

where $\theta \in [0, \pi]$, if $\theta$ is equal to 0, neither $\frac{\pi}{2}$ nor $\pi$ will update the dung beetle's position.

In dung beetle optimization algorithms, the choice of apt spawning sites by female dung beetles plays a pivotal role in ensuring the survival and procreation of their progeny. To model the behavior of female dung beetles when selecting a spawning area, we employ a boundary selection strategy as follows:

$$\begin{aligned} Lb^* &= \max(X^* \times (1 - R), Lb) \\ Ub^* &= \min(X^* \times (1 - R), Ub) \end{aligned} \tag{11}$$

here, $X^*$ represents the current local optimal position. The symbols $Lb^*$, and $Ub^*$ define the lower and upper boundaries of the spawning area, respectively. Furthermore, $R = 1 - t/T_{max}$, $T_{max}$ are maximum iteration numbers, while $Lb$ and $Ub$ specify the lower and upper constraints of the optimization problem. In the Dung Beetle Optimization Algorithm (DBO), upon establishing the spawning area, female dung beetles prioritize breeding balls within that vicinity for laying eggs. It's pivotal to highlight that every female dung beetle within the DBO framework produces a single breeding ball per iteration. The position of these breeding balls remains fluid throughout the iteration process, represented as:

$$B_i(t + 1) = X^* + b_1 \times (B_i(t) - Lb^*) + b_2 \times (B_i(t) - Ub^*) \tag{12}$$

in this context, the position of the $i$th breeding ball during the $t$th iteration is symbolized by $B_i(t)$, with $b_1$ and $b_2$ serving as two distinct random vectors, each of size $1 \times D$. Here, $D$ encapsulates the optimization problem's dimensionality. Importantly, the positioning of breeding balls adheres strictly to the confines of the designated spawning area. (Refer to Algorithm 2 for further details.)

**Algorithm 2** Breeding ball position update strategy

```
Input:
maximum number of iterations T_max, number of breeding balls N, current
number of iterations t
Output:
Location of the ith breeding ball B_i
R = 1 - t/T_max
for i ← 1 to n do
    Update the position of the breeding ball using Eq (12)
    for j ← 1 to D do
        if B_ij > Ub* then
```

```
            B_ij ← Ub*
        end if
        if B_ij < Lb* then
            B_ij ← Lb*
        end if
    end for
 end for
```

Fig 2 depicts the movement of rolling dung beetles, represented by dark blue dots, in a three-dimensional search space. The yellow dot at the center of a small sphere indicates the current local optimal position, $X^*$, representing the best solution found in the current iteration. Within this sphere, small black dots symbolize breeding balls, each enclosing a dung beetle egg. Red dots at the extremities of both the large and small spheres demarcate the upper and lower boundary limits. These boundaries restrict the beetles' rolling and egg-laying range, ensuring they search and reproduce effectively within the algorithm's optimal range.

Adult dung beetles, often referred to as 'baby dung beetles', emerge from the ground in search of food. To model the foraging behavior of dung beetles in their natural habitat, it's essential to define an optimal foraging area. The boundaries of this area are delineated as follows:

$$
\begin{aligned}
Lb^b &= \max(X^b \times (1 - R), Lb) \\
Ub^b &= \min(X^b \times (1 - R), Ub)
\end{aligned}
\tag{13}
$$

In this, $X^b$ represents the global optimal position while $Lb^b$ and $Ub^b$ respectively indicate the lower and upper bounds of the optimal foraging area. Further parameter definitions are given in Eq (11). Consequently, the position of the dung beetle is updated as:

$$
x_i(t + 1) = x_i(t) + C_1 \times (x_i(t) - Lb^b) + C_2 \times (x_i(t) - Ub^b)
\tag{14}
$$

here, $x_i(t)$ specifies the position of the $i$th dung beetle during the $t$th iteration. This update process involves two random vectors: $C_1$ and $C_2$. The former, $C_1$, is a random number following a normal distribution, aiding in modulating the exploratory behavior of the dung beetle. Meanwhile, $C_2$ is a random vector within the interval $(0, 1)$, adjusting the beetle's position in relation to both the globally optimal position and the optimal foraging area.

Additionally, a category of dung beetles, termed 'thieves', is integrated into the algorithm. Their primary role is to pilfer dung balls from fellow beetles for sustenance. As inferred from Eq (13), $X^b$ symbolizes the prime food source. It's plausible, then, to consider the vicinity of $X^b$ as the prime zone for food competition. As iterations proceed, the position data of these thieving dung beetles evolves and is characterized as follows:

$$
x_i(t + 1) = X^b + S \times g \times (|x_i(t) - X^*| + |x_i(t) - X^b|)
\tag{15}
$$

in this representation, $x_i(t)$ indicates the position of the $i$th thieving dung beetle at the $t$th iteration. Additionally, $g$ is a random vector of dimensions $1{\times}D$, adhering to a normal distribution, and $S$ is a constant.

Building upon the preceding discussion, the devised DBO algorithm first determines the maximum iteration count and sets the total population size of dung beetles as N. All agents are subsequently initialized at random, with their roles distributed based on a specified proportionate diagram. This distribution is visualized with sectors, where 20% corresponds to ball-rolling dung beetles, 20% to ball-breeding dung beetles, 25% to small dung beetles, and the remaining 35% to stealing dung beetles.

## 3D Boundary Selection Strategy Model

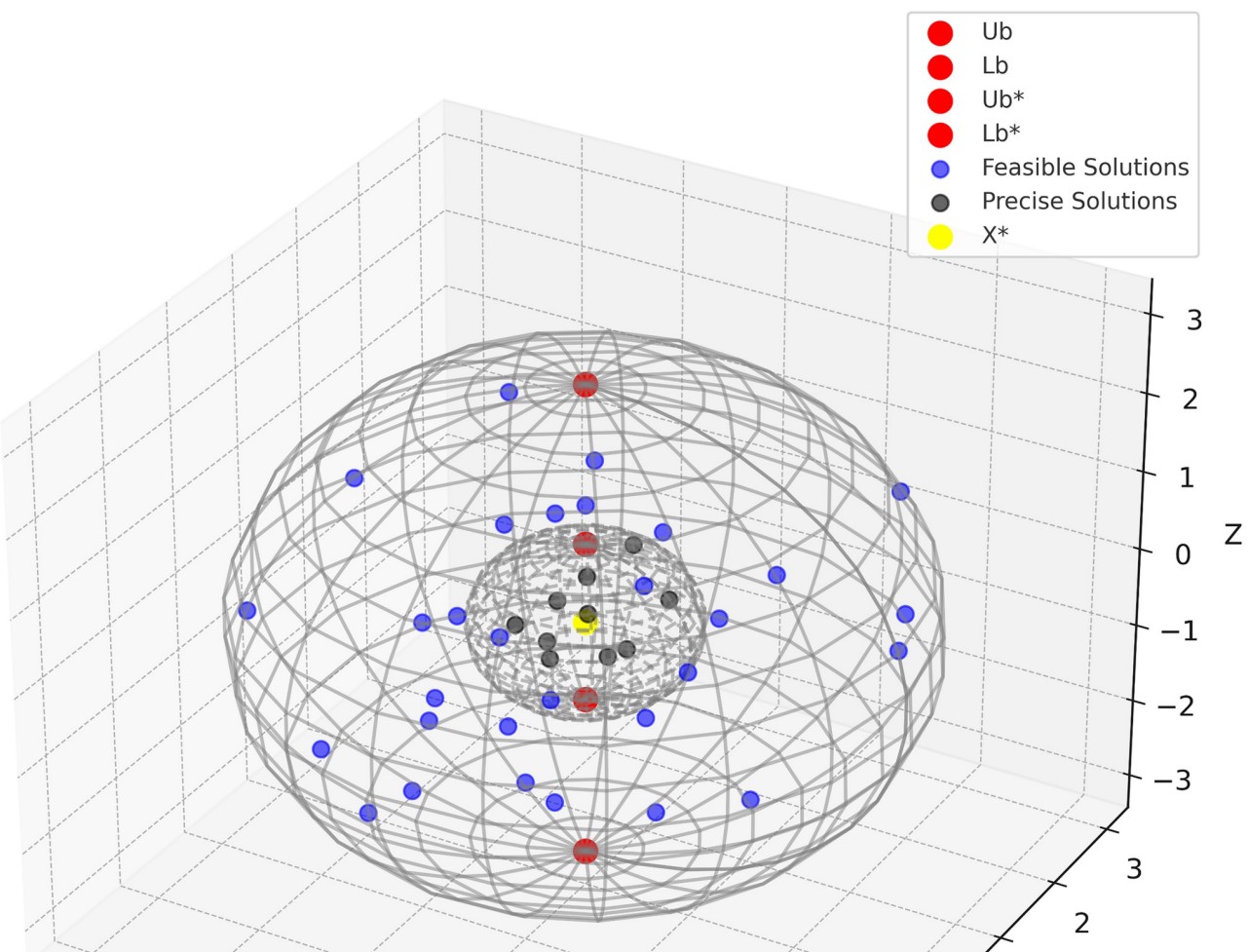

**Fig 2. The conceptual model of boundary selection strategy.**

For illustrative purposes, let's assume a total population of 30 dung beetles. Using Fig 3 as a guide, beetles are allocated to each agent category. Here, orange, yellow, green, and brown rectangles symbolize rolling dung beetles, breeding balls, small dung beetles, and stealing dung beetles respectively. This allocation ensures that during the algorithm's operation, dung beetles of distinct roles synergize based on their unique behaviors, aiming for enhanced optimization.

Subsequently, the positions of the rolling dung beetle, breeding ball, little dung beetle, and stealing dung beetle are incessantly refreshed. Guided by specific rules and equations within

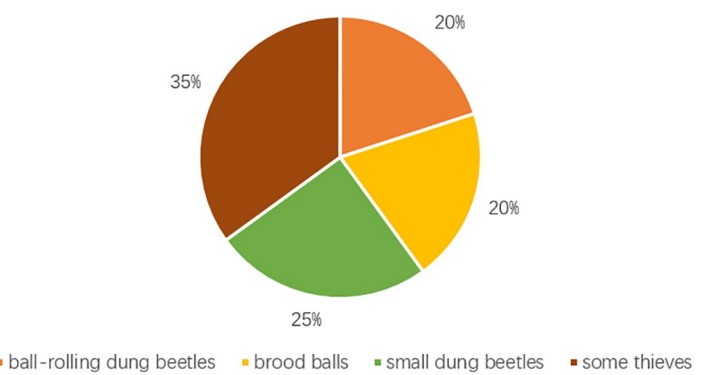

**Fig 3. The beetle proportion chart.**

the algorithm, they undergo adaptive shifts through iterative processes. Ultimately, the algorithm presents the global best position $X^b$, accompanied by its respective fitness value.

## 3 DBO-Otsu

### 3.1 Improvement of the traditional Otsu algorithm

A rubber cut mark map is randomly taken as in Fig 4, and its grayscale histogram is shown in Fig 5. In this experiment, the threshold value of the latex and cut mark region of interest is between 150–255, and it can be seen from the histogram that the traditional Otsu algorithm is affected by the global pixel distribution, and the threshold value (shown by the red solid line in the figure) will be to the left, which is affected by a large number of low grayscale regions, and it is unable to segment the region of our interest. If this threshold is used for segmentation, the segmentation map is shown in Fig 6, and it is obvious that it is impossible to distinguish the cut marks from the latex.

After experimental comparisons, the following improvements are proposed. The image of interest is represented by $L$ gray levels $(1; 2; \ldots; L)$. First, a suitable gray scale Th is set as the first threshold. According to the selected threshold, the gray scale of the image is divided into two parts:$[0, Th]$ and $[Th + 1, L − 1]$. For an image with pixels $N \times M$, the number of pixels with gray level $i$ is $n$, and the total number of pixels $n$ is

$$n = M \times N = \sum_{i=0}^{L-1} n_i \tag{16}$$

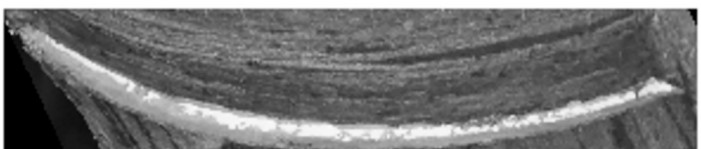

**Fig 4. Original image.**

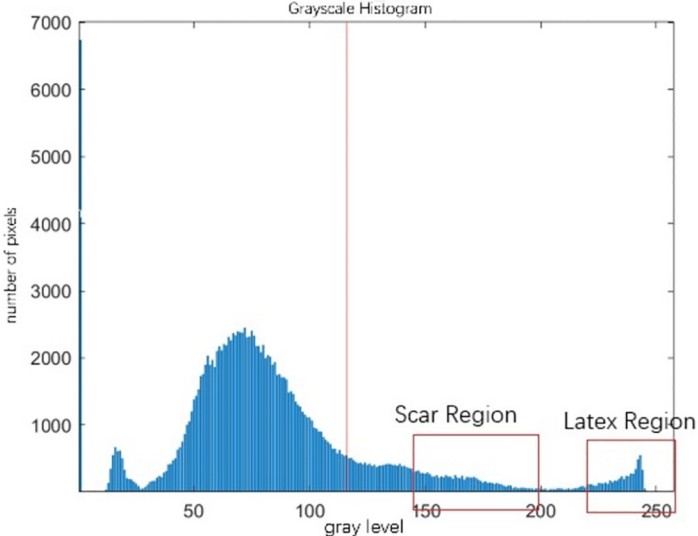

**Fig 5. Grayscale histogram.**

The number of pixels in $[0; Th]$ is $n_l$, and the number of pixels in $[Th + 1; L - 1]$ is $n_r$.

$$n_l = \sum_{i=0}^{Th} n_i \ \text{ and } \ n_r = \sum_{i=Th+1}^{L-1} n_i \tag{17}$$

The probability that a pixel is in $[0; Th]$ is $p_{il}$ and the probability that a pixel is in $[Th + 1; L - 1]$ is $p_{ir}$.

$$p_{il} = \frac{n_i}{n_l} \quad \text{and} \quad p_{ir} = \frac{n_i}{n_r} \tag{18}$$

Setting the gray values as $j, k, l, m$, the range of $[Th + 1; L - 1]$ is divided into five categories: $C_0, C_1, C_2, C_3$ and $C_4$. The distribution probability of $C_0, C_1, C_2, C_3$ and $C_4$ is $\omega_0, \omega_1, \omega_2, \omega_3$ and $\omega_4$,

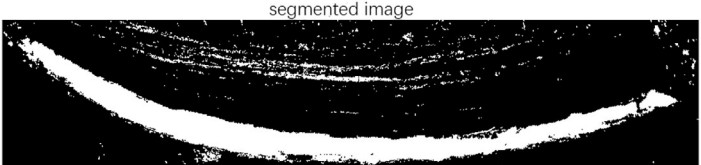

**Fig 6. Segmentation image.**

denoted as:

$$\omega_0 = \sum_{i=Th+1}^{j} p_{ir} \text{ and } \omega_1 = \sum_{i=j+1}^{k} p_{ir}$$

$$\omega_2 = \sum_{i=k+1}^{l} p_{ir} \text{ and } \omega_3 = \sum_{i=l+1}^{m} p_{ir} \tag{19}$$

$$\omega_4 = \sum_{i=m+1}^{L-1} p_{ir}$$

The average pixel gray probabilities of $C_0,C_1,C_2,C_3$ and $C_4$ are $\mu_0, \mu_1, \mu_2, \mu_3$ and $\mu_4$.

$$\mu_0 = \sum_{i=Th+1}^{j} i \cdot p_{ir} \text{ and } \mu_1 = \sum_{i=j+1}^{k} i \cdot p_{ir}$$

$$\mu_2 = \sum_{i=k+1}^{l} i \cdot p_{ir} \text{ and } \mu_3 = \sum_{i=l+1}^{m} i \cdot p_{ir} \tag{20}$$

$$\mu_4 = \sum_{i=m+1}^{L-1} p_{ir}$$

The average gray level $\mu$ in the range $[Th + 1; L - 1]$ can be expressed as follows:

$$\mu = \sum_{i=Th+1}^{L-1} i \cdot p_{ir} \tag{21}$$

The between-class variances for $C_0,C_1,C_2$ and $C_3$ were:

$$\sigma_B^2 = \omega_0(\mu_0 - \mu)^2 + \omega_1(\mu_1 - \mu)^2 \tag{22}$$

Referring to $j, k, l, m$, the optimal threshold are $j^*, k^*, l^*, m^*$ such that the maximum value is reached $\sigma_B^2(j^*, k^*, l^*, m^*)$

$$\sigma_B^2(j^*, k^*, l^*, m^*) = \max_{1 \leq j < k < l \leq L-1} \sigma_B^2(j, k, l, m) \tag{23}$$

The obtained $j^*, k^*, l^*, m^*$ range is in $[Th + 1; L - 1]$. The improved version processes $[Th + 1; L - 1]$ as separate images. As a result, the effect of a large number of pixels in the low gray range on the region of interest can be ignored in the selection of the threshold.

## 3.2 Otsu method improved with DBO algorithm

The time complexity of the improved Otsu method is $O(L^4)$. In order to reduce the computation time, we combine the DBO algorithm with the improved Otsu method and propose the DBO-Otsu method. The specific steps are as follows, as illustrated in Fig 7.

In the DBO-Otsu algorithm, the gray scale value K during Otsu calculation is considered as the coordinates X of the dung beetle population in the algorithm, and according to Eq (23) The fitness of each dung beetle individual is calculated, and the fitness is inverted. Then the DBO algorithm is used to simulate the behavioral patterns of dung beetles, comparing the fitness values and updating the coordinates $X$ in the iterative process, and finally finding the optimal threshold value to replace the exhaustive method in the traditional Otsu algorithm.

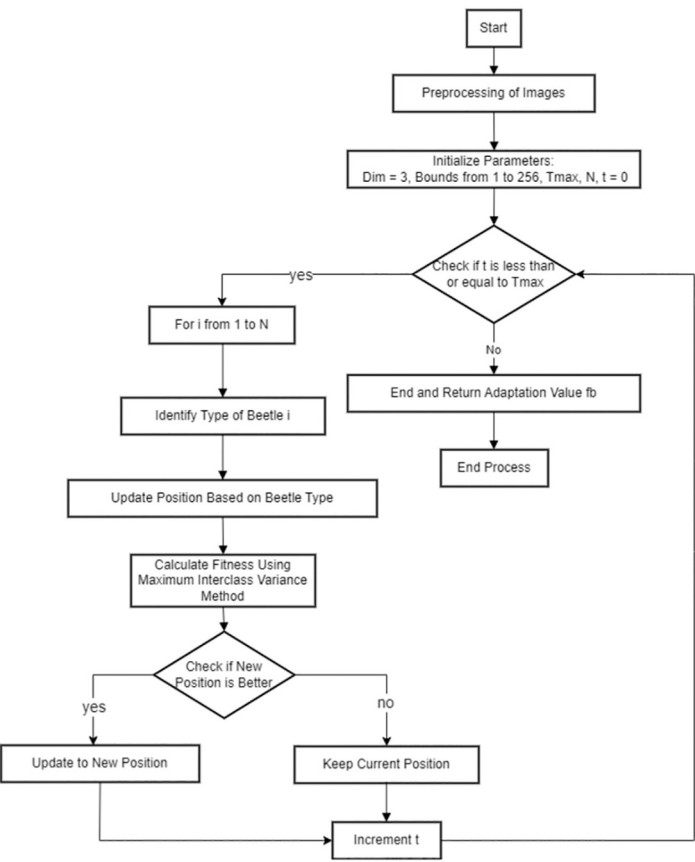

**Fig 7. DBO-Otsu flowchart.**

It should be noted that due to the design of the algorithm, the coordinates obtained result in floating point numbers, while the grayscale values of the image are in the discrete integer range [0, 255]. Therefore, when calculating the individual fitness, the coordinates need to be processed and limited to integers for subsequent calculations.

The DBO-Otsu pseudocode is shown in Algorithm 3.

**Algorithm 3** DBO-Otsu algorithm

```
Inputs: maximum number of iterations Tmax, population size N
Outputs: optimal position Xb and its fitness value fb
Randomly initialize the dung beetle population i ← 1, 2, ..., N
Initialize parameters: Dim = 4, bounds ∈ [1, 256], T = 150, t = 0,
N = 60
while t ≤ Tmax
  for i ← 1 to N do
    if i = = Dung Beetle then
      δ = rand(1)
      if δ < 0.9 then
        Use Algorithm 1 to select α
        Update location using formula (9)
      else
        Update location using Eq (12)
      end if
    else if i = = Breeding Balls then
```

```
          Update using Algorithm 2
      else if i == Little Dung Beetle then
          Update using Eq (14)
      else if i == Stealing Dung Beetles then
          Update using formula (15)
      end if
    end for
    if new position is better then
        Update it
    end if
    t = t + 1
  end while
  return Adaptation value f_b
```

### 3.3 Segmentation strategies of DBO-Otsu at different levels

In low-level Tapping Panel Dryness (TPD) images, where the latex quantity has not significantly diminished, as illustrated in Figs 8–10.

The final threshold in the multi-threshold output of the DBO-Otsu algorithm provides a high-quality segmentation of the latex. However, for images of higher-level TPD, where latex is sparse, as shown in Figs 11–13, the original approach, resulting in Figs 14–16, often fails to reflect the actual scenario. In these cases, the latex regions no longer manifest as distinct peaks on the grayscale histogram, rendering traditional multi-threshold segmentation methods ineffective in isolating the latex areas.

To address this challenge, this study introduces an improved segmentation strategy, specifically for high-level TPD images with scarce latex. This method initially performs multi-threshold segmentation using DBO-Otsu, followed by a morphological assessment of the tapping cut images to examine the segmentation outcome. If the segmented shape is not curvilinear, it is

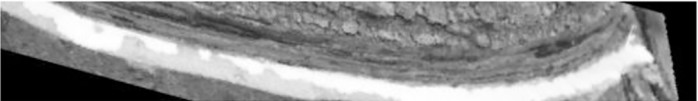

**Fig 8. Low-level epidermis disease Fig 1.**

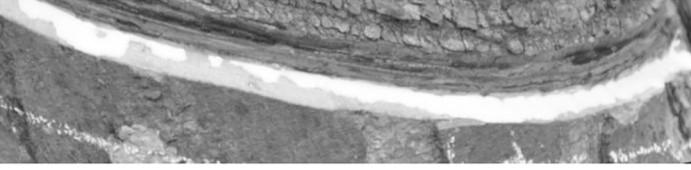

**Fig 9. Low-level epidermis disease Fig 2.**

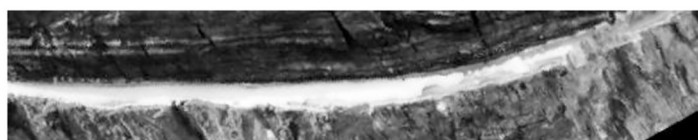

**Fig 10. Low-level epidermis disease Fig 3.**

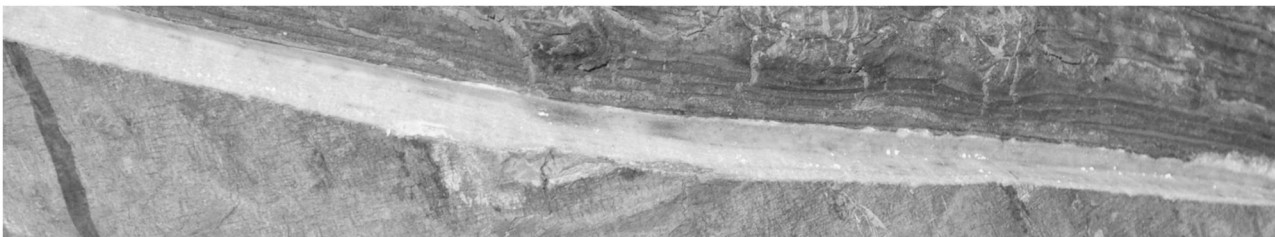

**Fig 11. High-level epidermis disease Fig 1.**

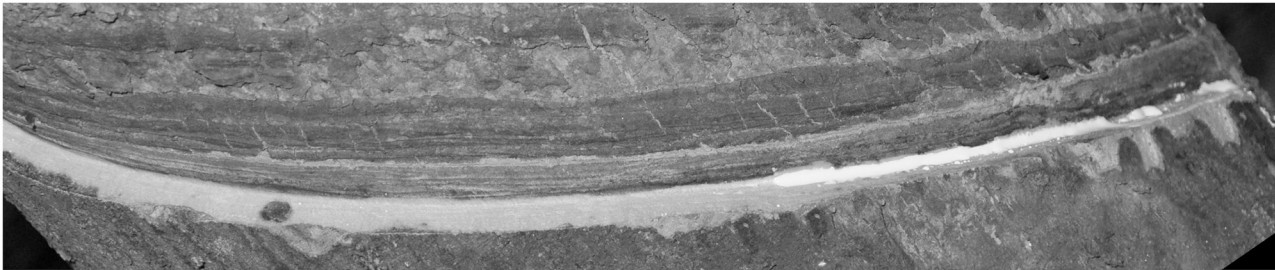

**Fig 12. High-level epidermis disease Fig 2.**

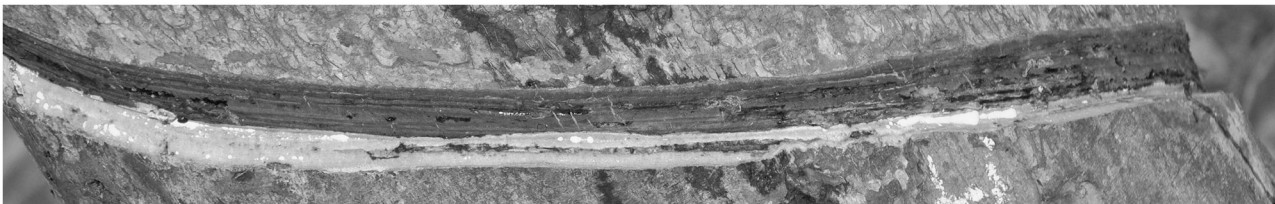

**Fig 13. High-level epidermis disease Fig 3.**

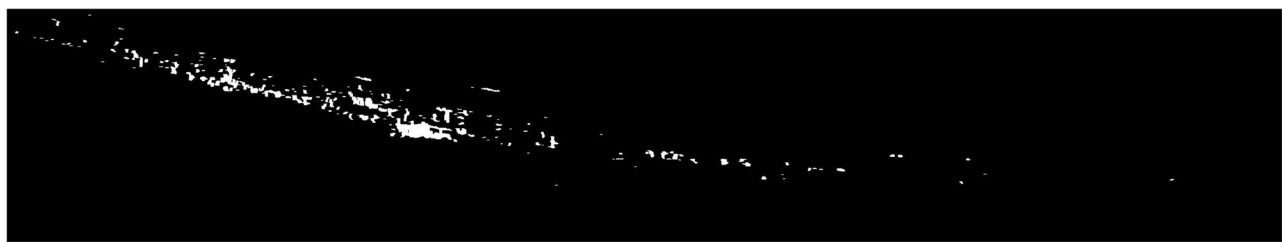

**Fig 14. High-level epidermis disease Latex Fig 1.**

identified as a high-level TPD image. The strategy then utilizes the maximum non-zero value at the end of the grayscale histogram as the final latex segmentation threshold, thereby precisely locating the latex areas. This approach considers the high grayscale value but low pixel count characteristic of the latex, enhancing accuracy in identifying and segmenting sparse

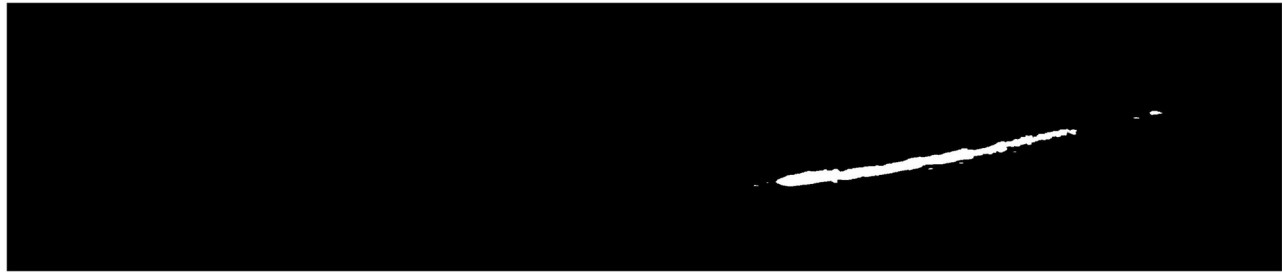

**Fig 15. High-level epidermis disease Latex Fig 2.**

latex regions. Moreover, it effectively avoids missegmentation due to histogram noise or high grayscale values in non-latex areas, crucial for analyzing high-level TPD images as accurate extraction of latex areas is vital for disease assessment and subsequent processing. Re-segmenting the latex using this method, as represented in Figs 14–16, aligns the results with actual conditions, successfully extracting the latex images.

## 4 Experiments and analysis of results

### 4.1 Parameter settings of the algorithm

In order to assess the performance of the proposed DBO-Otsu algorithm, we randomly selected three images from the rubber dataset as benchmark images. Due to the stochastic nature of metaheuristic algorithms, the results vary with each execution. In this context, each algorithm was subjected to 50 experimental trials, and the results were then averaged. This method was compared for performance with the original Otsu method, SSA-Otsu method [80], WOA-Otsu method [81], WSO-Otsu method [82], GWO-Otsu method [83], AHA-Otsu method [84], and CSA-Otsu method [85]. The parameter settings for each algorithm are shown in Table 1. Except for pop_num and Max_iter, which were modified to accommodate the complexity of the experiments, the regional parameters were adopted from recommended studies.

All these experiments were conducted using MATLAB R2022b on a Windows 11 operating system, with 16GB RAM memory and an Intel Core i5–11300 H CPU operating at 3.10 GHz.

### 4.2 Evaluation metrics

**4.2.1 PSNR.** Peak Signal-to-Noise Ratio (PSNR) is a widely used metric in image and video processing for objective quality assessment. It is defined as the ratio between the maximum possible power of a signal and the power of the noise that affects the fidelity of its representation. The PSNR is usually expressed in logarithmic decibel scale. The PSNR is calculated

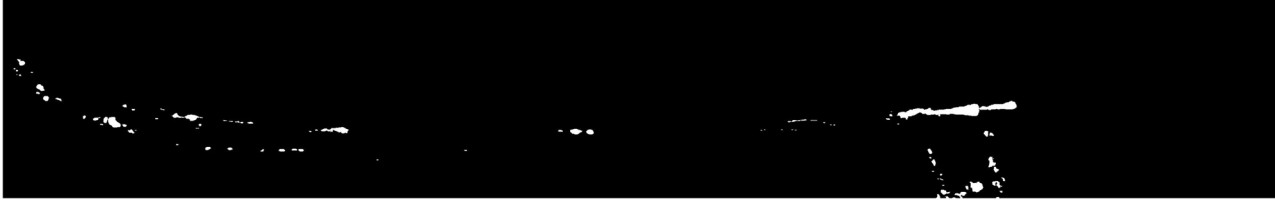

**Fig 16. High-level epidermis disease Latex Fig 3.**

**Table 1. Parameter setting of the testing algorithm.**

| Algorithm | Parameter | Setting |
|---|---|---|
| SSA-Otsu | pop_num | 60 |
| | Max_iter | 200 |
| WOA-Otsu | pop_num | 60 |
| | Max_iter | 200 |
| WSO-Otsu | pop_num | 60 |
| | Max_iter | 200 |
| | fmax | 0.75 |
| | fmin | 0.07 |
| | tau | 4.11 |
| | pmin | 0.5 |
| | pmax | 1.5 |
| | a0 | 6.25 |
| | a1 | 100 |
| | a2 | 0.0005 |
| GWO-Otsu | pop_num | 60 |
| | Max_iter | 200 |
| AHA-Otsu | pop_num | 60 |
| | Max_iter | 50 |
| CSA-Otsu | pop_num | 60 |
| | Max_iter | 200 |
| | rho | 1 |
| | p1 | 2 |
| | p2 | 2 |
| | c1 | 2 |
| | c2 | 1.8 |
| | gamma | 2 |
| | alpha | 4 |
| | beta | 3 |
| DBO-Otsu | pop_num | 60 |
| | Max_iter | 200 |
| | P_percent | 0.2 |
| | k | 0.1 |
| | b | 0.3 |
| | S | 0.5 |

using the following formula:

$$\text{PSNR} = 10 \cdot \log_{10}\left(\frac{MAX_I^2}{MSE}\right) \qquad (24)$$

Where: $MAX_I$ is the maximum possible pixel value of the image. $MSE$ is the Mean Squared Error between the reference image and the distorted image.

**4.2.2 FSIM.** Feature Similarity Index (FSIM) is a more advanced metric that considers luminance, contrast, and structure similarities between the reference and distorted images to compute the similarity index. The FSIM is calculated using the following formula:

$$\text{FSIM} = \frac{1}{N}\sum \frac{2 \cdot \mu_X \cdot \mu_Y + C_1}{\mu_X^2 + \mu_Y^2 + C_1} \cdot \frac{2 \cdot \sigma_{XY} + C_2}{\sigma_X^2 + \sigma_Y^2 + C_2} \qquad (25)$$

Where: $\mu_X$ and $\mu_Y$ are the local means of images $X$ and $Y$, respectively. $\sigma_X$ and $\sigma_Y$ are the local standard deviations of images $X$ and $Y$, respectively. $\sigma_{XY}$ is the local cross covariance between images $X$ and $Y$. $C_1$ and $C_2$ are constants. $N$ is the total number of pixels in the images.

**4.2.3 SSIM.** Structural Similarity Index (SSIM) is another advanced metric for comparing the similarity between two images. The SSIM index is designed to improve on traditional metrics like PSNR and MSE by considering changes in structural information, luminance, and contrast. The SSIM is calculated using the following formula:

$$\text{SSIM}(x, y) = \frac{(2 \cdot \mu_x \cdot \mu_y + C_1)}{(\mu_x^2 + \mu_y^2 + C_1)} \cdot \frac{(2 \cdot \sigma_{xy} + C_2)}{(\sigma_x^2 + \sigma_y^2 + C_2)} \tag{26}$$

Where: $\mu_x$ and $\mu_y$ are the mean of images $x$ and $y$, respectively. $\sigma_x$ and $\sigma_y$ are the variance of images $x$ and $y$, respectively. $\sigma_{xy}$ is the covariance of images $x$ and $y$. $C_1$ and $C_2$ are constants used to avoid division by zero.

## 4.3 Indicator testing

In our comparative study, the DBO-Otsu algorithm was evaluated against six other advanced Otsu methodologies: SSA-Otsu, WOA-Otsu, WSO-Otsu, GWO-Otsu, AHA-Otsu, and CSA-Otsu. We employed several metrics for this assessment, including runtime, PSNR, FSIM, and SSIM. Runtime, as a measure of real-time performance, is a critical factor in gauging the efficiency of an algorithm. PSNR, FSIM, and SSIM, which are closely tied to the structural attributes of images, serve as indicators of the segmentation quality. Selected results from this comparative analysis are shown in Table 2, focusing on operational data for levels 4–6, 4–12, 4–19, and 4–20.

In terms of runtime, the DBO-Otsu algorithm demonstrated superior performance over other enhanced Otsu methods, maintaining moderate processing times in all experimental setups. In the analysis of the DBO-Otsu method, a comprehensive evaluation was conducted, focusing on the average PSNR, FSIM, and SSIM scores under diverse experimental conditions.

The assessment in the Table 3 revealed that the DBO-Otsu method consistently demonstrated high performance. Specifically, it achieved the highest average ranking in both PSNR and SSIM scores, with an impressive average rank of 1.50 for each. In the FSIM category, DBO-Otsu also performed commendably, securing an average rank of 2.25. These rankings underscore its proficiency in several key areas, particularly in one of the experimental domains where it excelled. The integration of the DBO-Otsu method with the advanced DBO algorithm has been instrumental in enhancing segmentation accuracy.

The statistical significance of DBO-Otsu's performance was determined using a Wilcoxon signed-rank test at a significance level of 0.1. The outcomes in Table 4 revealed statistically significant results for the DBO-Otsu method in terms of PSNR and SSIM, whereas the FSIM scores did not reach a similar level of statistical significance. These results confirm the effectiveness of the DBO-Otsu method and highlight areas for potential refinement.

To visually illustrate the efficacy of various optimization algorithms in threshold optimization, we generated convergence curves, as depicted in Fig 17. The horizontal axis on these curves represents the number of iterations, while the vertical axis reflects the best fitness value achieved to date.

As can be seen in Fig 17, although other algorithms like SSA demonstrated superior final results in some experiments (such as 4–8, 4–19, and 4–20), DBO-Otsu exhibited strong performance in both convergence speed and accuracy, which was particularly evident in most of the tested functions.

**Table 2. Test result of seven algorithms.**

| Group | Algorithm | ScarThreshold | LatexThreshold | Time | PSNR | FSIM | SSIM |
|---|---|---|---|---|---|---|---|
| 4–6 | SSA | 170.607846 | 202.3673229 | 0.454 | 8.149481674 | 0.514599926 | 0.126894101 |
| | WOA | 193.3105468 | 255.4191533 | 0.292 | 7.569149399 | 0.505307484 | 0.114336442 |
| | WSO | 174.129266 | 202.9681429 | 0.293 | 8.067329821 | 0.513251516 | 0.123459772 |
| | GWO | 170.9923696 | 202.4617205 | 0.297 | 8.149481674 | 0.514599926 | 0.126894101 |
| | AHA | 170.6434785 | 202.960859 | 0.326 | 8.149481674 | 0.514599926 | 0.126894101 |
| | CSA | 170.704867 | 202.3591813 | 0.293 | 8.149481674 | 0.514599926 | 0.126894101 |
| | DBO | 171.4228188 | 204.8375643 | **0.372** | **8.165899285** | **0.521412398** | **0.127876046** |
| 4–12 | SSA | 155.5854386 | 207.8677594 | 0.44 | 8.289015524 | 0.557163498 | 0.09489891 |
| | WOA | 149.0694241 | 203.5416501 | 0.293 | **15.17593885** | **0.590534328** | 0.550520428 |
| | WSO | 154.2989164 | 202.5409691 | **0.283** | 8.333545453 | 0.558486259 | 0.100172884 |
| | GWO | 157.7155776 | 205.8915754 | 0.3 | 8.224993212 | 0.556169337 | 0.093358665 |
| | AHA | 155.2227337 | 207.6226922 | 0.332 | 8.289015524 | 0.557163498 | 0.09489891 |
| | CSA | 155.304461 | 207.5493689 | 0.297 | 8.289015524 | 0.557163498 | 0.09489891 |
| | DBO | 152.3887825 | 204.0333982 | 0.335 | 8.364000908 | 0.557133382 | **0.100182992** |
| 4–19 | SSA | 171.4507 | 193.7683 | 0.45 | 11.44445 | 0.444172 | 0.074768 |
| | WOA | 171.3533 | 193.0518 | 0.357 | 11.37516 | 0.432095 | 0.071192 |
| | WSO | 166.0914 | 195.5994 | **0.338** | 11.49231 | 0.440329 | 0.076701 |
| | GWO | 171.3085 | 193.8558 | 0.931 | 11.44445 | 0.444172 | 0.074768 |
| | AHA | 171.8939 | 193.1976 | 0.596 | 11.44445 | 0.444172 | 0.074768 |
| | CSA | 168.8158 | 193.9061 | 0.366 | 11.51899 | 0.452194 | 0.078359 |
| | DBO | 166.6904 | 193.8078 | 0.462 | **11.54439** | **0.452779** | **0.079371** |
| 4–20 | SSA | 197.4308 | 209.5352 | 0.601 | 6.124981 | 0.60167 | 0.041313 |
| | WOA | 182.7351 | 208.6473 | 0.4 | **11.72238** | 0.613376 | **0.562652** |
| | WSO | 193.5352 | 213.5405 | 0.871 | 6.128787 | 0.606108 | 0.045566 |
| | GWO | 197.0394 | 209.767 | 0.491 | 6.124981 | 0.60167 | 0.041313 |
| | AHA | 197.4107 | 209.7839 | 0.405 | 6.124981 | 0.60167 | 0.041313 |
| | CSA | 197.0325 | 209.6747 | **0.361** | 6.124981 | 0.60167 | 0.041313 |
| | DBO | 181.2085 | 208.6152 | 0.464 | 7.256473 | **0.641461** | 0.133997 |

Note: Bold indicates the best score for each item

## 4.4 Detail verification

In Table 1, we present the selected thresholds and describe the division of the image into five regions based on these thresholds, with each pixel's value determined by its corresponding region. The detailed assessment process includes: firstly applying multi-threshold processing on the original image using different methods, then selecting regions containing key

**Table 3. Final average rankings of algorithms in PSNR, FSIM, and SSIM.**

| Algorithm | Average PSNR Ranking | Average FSIM Ranking | Average SSIM Ranking |
|---|---|---|---|
| SSA | 4.75 | 4.25 | 4.75 |
| WOA | 4.00 | 4.25 | 4.00 |
| WSO | 3.75 | 4.25 | 3.75 |
| GWO | 5.25 | 5.00 | 5.25 |
| AHA | 4.75 | 4.25 | 4.75 |
| CSA | 4.00 | 3.75 | 4.00 |
| DBO | 1.50 | 2.25 | 1.50 |

**Table 4. Wilcoxon test results comparing DBO with other algorithms.**

| Algorithm | PSNR p-value | FSIM p-value | SSIM p-value |
|---|---|---|---|
| SSA | 0.0625 | 0.1250 | 0.0625 |
| WOA | 0.8125 | 0.4375 | 0.8125 |
| WSO | 0.0625 | 0.1250 | 0.0625 |
| GWO | 0.0625 | 0.0625 | 0.0625 |
| AHA | 0.0625 | 0.1250 | 0.0625 |
| CSA | 0.0625 | 0.1250 | 0.0625 |

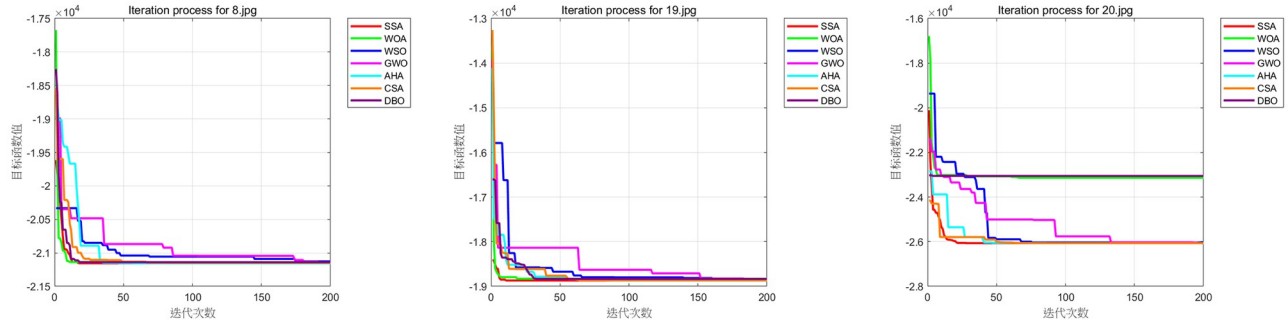

**Fig 17. Iterative comparison figure.**

information for comparison. Lastly, we calculate the difference in grayscale values for each pixel in these regions compared to the corresponding areas in the original image to determine a detail score.

To more clearly demonstrate and evaluate these details, specific regions were analyzed in Figs 18–20, which contain detailed information about the rubber tree tapping cuts (see Fig 21).

Furthermore, we established a scoring system for curve similarity as follows:

$$\text{Score} = \sum_{i=1}^{L} \text{score}(i)$$

$$score(i) = \begin{cases} 1, |gray(i) - gray1(i)| \le 50 \\ -1, |gray(i) - gray1(i)| > 50 \end{cases} \quad (27)$$

Where $i$ is the index of the pixel in the selected region, $L$ is the total number of pixels in that region, $gray1(i)$ represents the grayscale value of the processed curve at the $i$th point, and $gray(i)$ is the grayscale value of the original curve at the $i$th point.

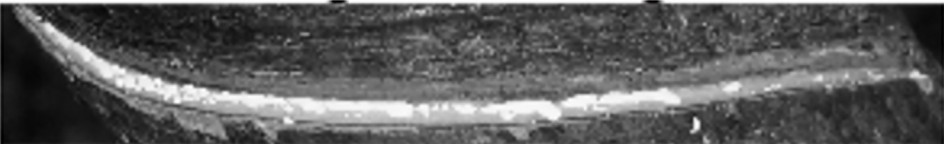

**Fig 18. Detail analysis of original image 1.**

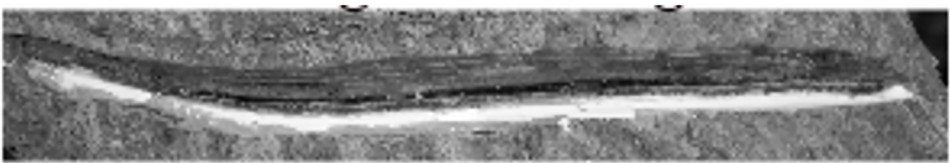

**Fig 19. Detail analysis of original image 2.**

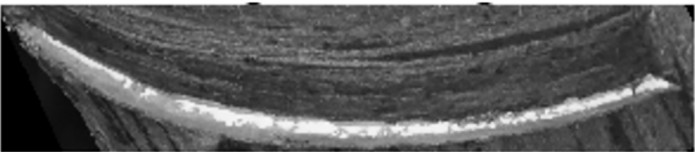

**Fig 20. Detail analysis of original image 3.**

The score increases when the difference in values at a given point between the two curves is small; it decreases when the difference is large. The total scores are then summed to obtain a final score. By comparing the scores in Table 3, we detailed the segmentation results of various algorithms.

The score increases when the difference in values at a given point between the two curves is small; it decreases when the difference is large. The total scores are then summed to obtain a final score. By comparing the scores in Table 5, we detailed the segmentation results of various algorithms.

In each set of experiments, DBO-Otsu scored the highest in detail retention, demonstrating its superiority in preserving original image details compared to other algorithms.

## 4.5 Application evaluation

To ascertain the DBO-Otsu algorithm's practical effectiveness developed in this research, we conducted segmentation tests using images from each level of the dataset, as illustrated in Fig 22.

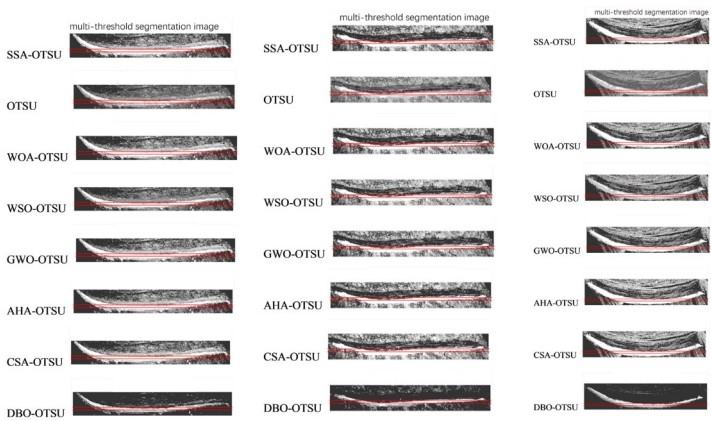

**Fig 21. Comparison of detail analysis.**

**Table 5. Fitness score table.**

| Algorithm | A 3–8 | B 3–7 | C 1–3 | Mean |
|---|---|---|---|---|
| SSA-Otsu | -816 | 960 | 186 | 110 |
| Otsu | -414 | 1944 | 846 | 792 |
| WOA-Otsu | -816 | 972 | 234 | 130 |
| WSO-Otsu | -600 | 522 | 468 | 130 |
| GWO-Otsu | -822 | 1002 | 474 | 218 |
| AHA-Otsu | -840 | 1002 | 60 | 74 |
| CSA-Otsu | -750 | 858 | 102 | 70 |
| DBO-Otsu | 636 | 1248 | 2874 | 1586 |

The segmentation of tapping cuts, excluding those in level 3 images, proved effective in the remaining images, as evidenced in Fig 23. The level 3 images, characterized by blurred feature boundaries, presented a challenge, where the automated segmentation approach may not have been entirely suitable, resulting in less than optimal outcomes.

Furthermore, latex segmentation was executed on the original images, yielding the results shown in Fig 24. These results demonstrate that the DBO-Otsu algorithm successfully segments images even at levels 4 and 5, where latex pixels are sparse, thus overcoming the traditional Otsu method's limitations in handling areas with scant grayscale pixels. In the case of level 1, 2, and 3 images, some missegmentation occurred. However, these inaccuracies were addressed in the final statistical analysis, ensuring the overall results remained within an acceptable margin.

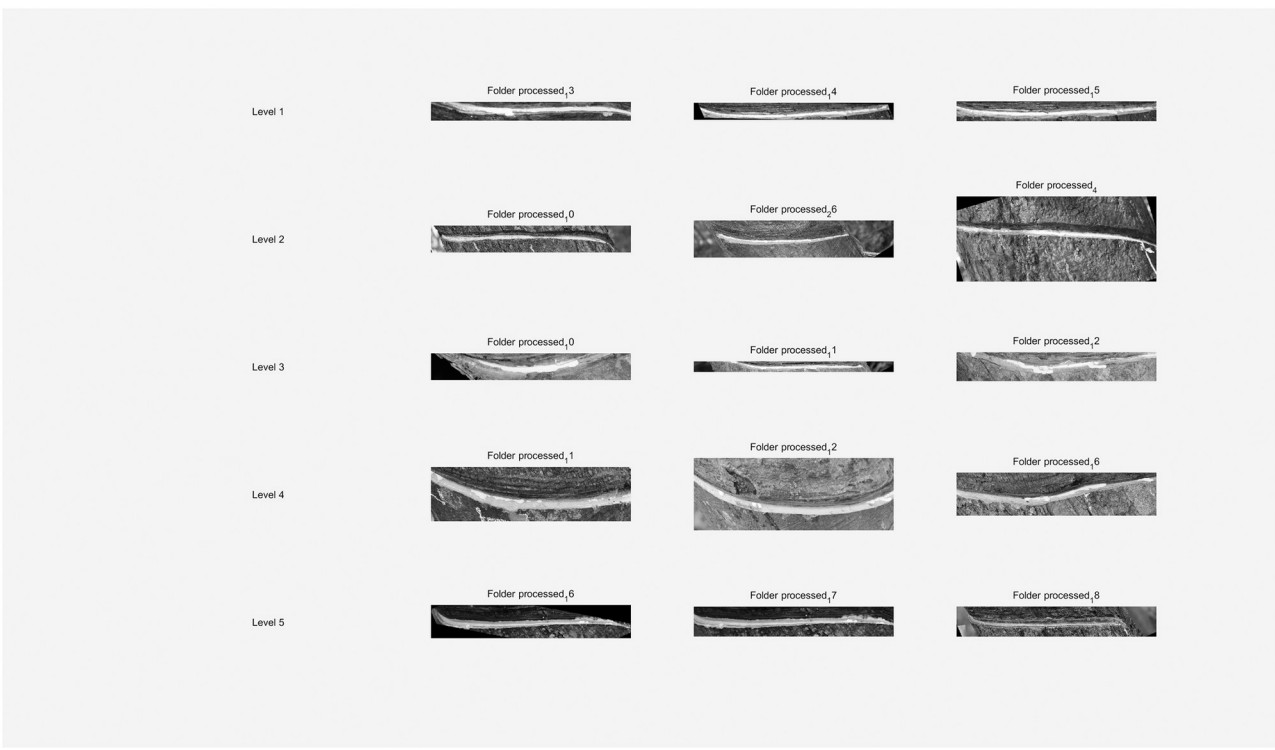

**Fig 22. Application evaluation of the original image.**

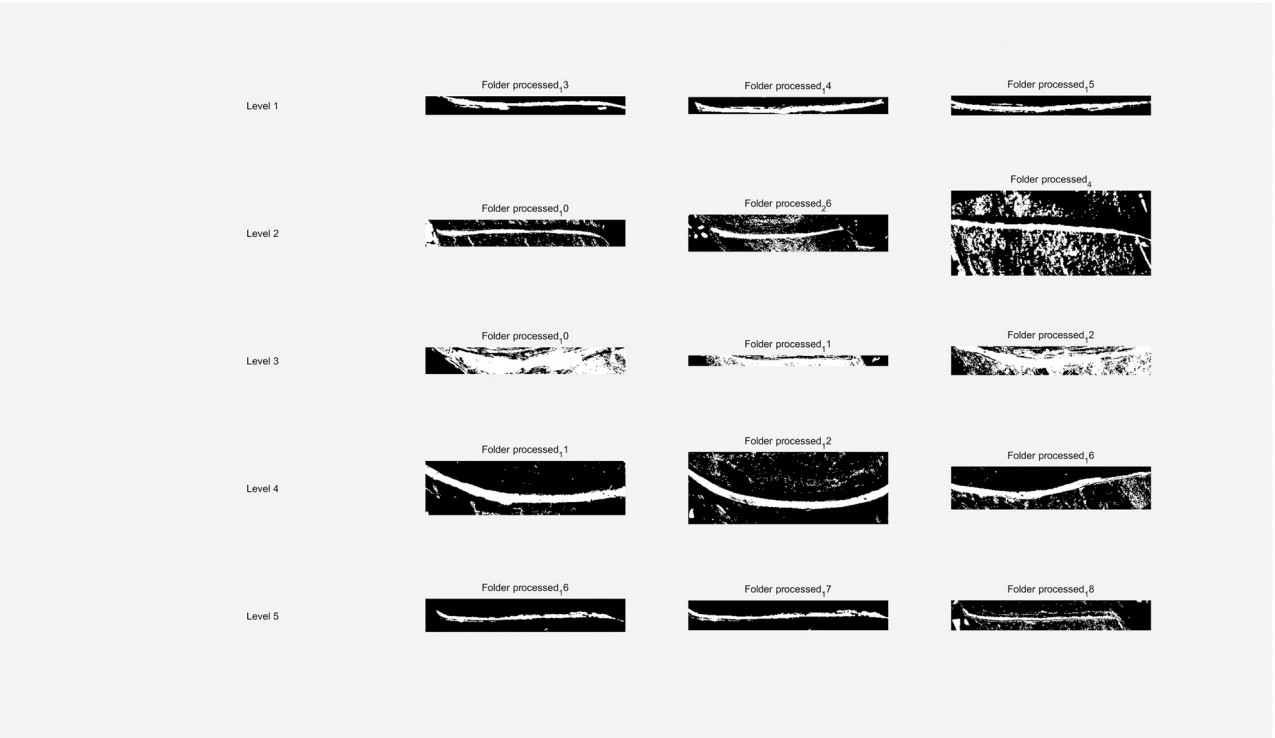

**Fig 23. Application evaluation of the incision image.**

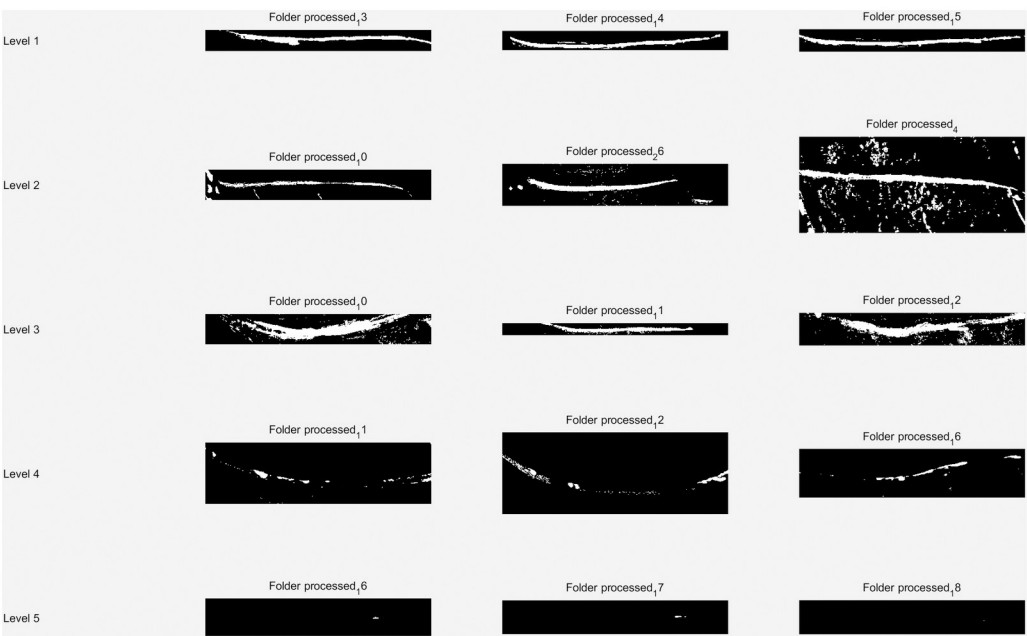

**Fig 24. Application evaluation of the latex image.**

**Table 6. TPD grade determination table.**

| Latex/Incision Ratio ($\lambda$) | Determination Grade |
|---|---|
| 1% $\leq \lambda \leq$ 10% | Grade 5 |
| 10% $< \lambda \leq$ 20% | Grade 4 |
| 20% $< \lambda \leq$ 35% | Grade 3 |
| 35% $< \lambda \leq$ 45% | Grade 2 |
| $\lambda >$ 45% | Grade 1 |

As defined in the criteria for calculating TPD levels in the reference [86], the ratio of latex to tapping cuts is designated as $\lambda = S_1/S_2$, where $S_1$ represents the area of latex and $S_2$ denotes the area of tapping cuts. Post-segmentation, the black areas in the images represent the background, while the white areas signify the regions of interest. The ratio of $\lambda$ can be approximated by calculating the proportion of pixels covering both the tapping cuts and the latex. The criteria for classifying different levels of TPD in this study are detailed in Table 6.

Utilizing MATLAB software, we quantified the number of pixels pertaining to tapping cuts and latex in Figs 23 and 24. This analysis enabled the determination of TPD levels based on their ratio, as depicted in Table 7.

The algorithm demonstrated exceptional proficiency in segmenting both latex and tapping cuts. It precisely identified the TPD levels through the calculated ratios, offering accuracy and efficiency that surpass traditional manual identification methods, thereby significantly reducing labor costs.

Additionally, we conducted a random evaluation of images across different TPD levels within the dataset, analyzing 15–20 images per level. The findings of this evaluation are summarized in the Table 8:

These results reveal that the segmentation performance of this method in level 1–2 and 4–5 images is markedly superior to that in level 3 images, with an accuracy rate exceeding 80%. The diminished accuracy observed in level 3 images is attributed to the nuances in segmentation accuracy. Higher-level image segmentation accuracy declines with an increasing area

**Table 7. Level determination table.**

| levels | Image Number | Latex Pixel Count | Scar Pixel Count | Area Ratio | Disease Level |
|---|---|---|---|---|---|
| level 1 | 13 | 7681 | 11099 | 0.692044328 | 1 |
| | 14 | 13842 | 24797 | 0.558212687 | 1 |
| | 15 | 11545 | 18598 | 0.620765674 | 1 |
| level 2 | 4 | 41651 | 96132 | 0.433268839 | 2 |
| | 10 | 182369 | 492541 | 0.370261562 | 2 |
| | 26 | 263777 | 612148 | 0.430903964 | 2 |
| level 3 | 10 | 175705 | 547090 | 0.32116288 | 3 |
| | 11 | 173405 | 590372 | 0.293721586 | 3 |
| | 12 | 147825 | 505293 | 0.292553034 | 3 |
| level 4 | 11 | 15753 | 194899 | 0.080826479 | 5 |
| | 12 | 60878 | 525198 | 0.115914379 | 4 |
| | 16 | 19962 | 193855 | 0.102973872 | 4 |
| level 5 | 16 | 126 | 12723 | 0.009903325 | 5 |
| | 17 | 128 | 11821 | 0.010828187 | 5 |
| | 18 | 189 | 399566 | 0.000473013 | 5 |

**Table 8. Accuracy rate table.**

| Judgment Level | Image Count | Accuracy Rate | Average Accuracy Rate |
|:---:|:---:|:---:|:---:|
| 1 | 26 | 92% | 84.43% |
| 2 | 25 | 92% | |
| 3 | 20 | 65% | |
| 4 | 20 | 81% | |
| 5 | 20 | 88% | |

ratio, whereas for lower-level images, it decreases with a decreasing area ratio. This results in a moderate latex-to-tapping cut ratio for level 3 images, posing a challenge for accurate segmentation identification.

## 5 Conclusions

The primary aim of this study was to enhance image segmentation by refining the classic Otsu thresholding method, with a specific focus on preserving intricate details. The motivation behind this research was to address the challenge of exponential time complexity growth in multi-level threshold computations. To achieve this, we introduced the innovative DBO algorithm, which was integrated into the Otsu method to create the DBO-Otsu algorithm—a novel image segmentation tool.

Our rigorous performance evaluation of the DBO-Otsu algorithm encompassed a comprehensive set of performance metrics, including PSNR, FSIM, and SSIM. The results demonstrated that DBO-Otsu not only maintained computational efficiency but also significantly reduced image distortion. In fact, DBO-Otsu surpassed the performance of six other comparative methods in preserving image structural integrity.

In practical applications of image segmentation, we encountered variations in latex quantities across different rubber tree disease levels. It became evident that a direct application of DBO-Otsu might not suffice for all scenarios. Therefore, we adopted a nuanced approach by conducting morphological analyses post-initial segmentation and adapting strategies tailored to images at various disease stages. While accuracy experienced a slight decline in images of intermediate disease levels, the majority of judgments remained acceptably accurate, with minimal errors.

In conclusion, our findings underscore the importance of prioritizing the DBO-Otsu algorithm in future research endeavors, especially in contexts where rapid and efficient TPD diagnosis is paramount. Notably, in instances with pronounced disease symptoms, the DBO-Otsu algorithm has the potential to deliver even more remarkable results. This approach not only expedites computation but also upholds high image quality, presenting a robust and efficient solution in image segmentation. However, it is crucial to acknowledge the inherent limitations within the algorithm, which excel in diagnosis based on relative proportions but may still face challenges in isolating specific targets. Our future research will be dedicated to enhancing segmentation accuracy, including the potential incorporation of edge detection algorithms to eliminate irrelevant areas on the trunk.

## Supporting information

**S1 Data.**
(ZIP)

**S2 Data.**

(ZIP)

**S3 Data.**

(ZIP)

**S4 Data.**

(ZIP)

**S5 Data.**

(ZIP)

**S6 Data.**

(ZIP)

**S7 Data.**

(ZIP)

**S8 Data.**

(ZIP)

**S9 Data.**

(ZIP)

**S10 Data.**

(ZIP)

**S1 File.**

(ZIP)

**S2 File.**

(ZIP)

## Author Contributions

**Formal analysis:** Zhenjing Xie.

**Funding acquisition:** Yongna Liu.

**Investigation:** Zhenjing Xie, Yongna Liu.

**Methodology:** Zhenjing Xie, Yongna Liu.

**Resources:** Yongna Liu.

**Supervision:** Weirui Tang, Yongna Liu.

**Writing – original draft:** Zhenjing Xie, Jinran Wu.

**Writing – review & editing:** Zhenjing Xie, Jinran Wu, Weirui Tang.

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
