## [Decision Letter · Decision Letter 0]

2 Nov 2023

PONE-D-23-30775Advancing Image Segmentation with DBO-Otsu: Addressing Rubber Tree Diseases through Enhanced Threshold TechniquesPLOS ONE

Dear Dr. Liu,

Thank you for submitting your manuscript to PLOS ONE. After careful consideration, we feel that it has merit but does not fully meet PLOS ONE’s publication criteria as it currently stands. Therefore, we invite you to submit a revised version of the manuscript that addresses the points raised during the review process.

We look forward to receiving your revised manuscript.

Kind regards,

Khan Bahadar Khan, Ph.D

Academic Editor

PLOS ONE

Journal Requirements:

"Hainan Provincial Natural Science Foundation of China (623RC449)"  

6. Please ensure that you refer to Figure 1, 2, 12, 13 and 14 in your text as, if accepted, production will need this reference to link the reader to the figure.

Reviewers' comments:

Reviewer's Responses to Questions

**Comments to the Author**

1. Is the manuscript technically sound, and do the data support the conclusions?

Reviewer #1: Yes

Reviewer #2: Partly

2. Has the statistical analysis been performed appropriately and rigorously? 

Reviewer #1: Yes

Reviewer #2: No

3. Have the authors made all data underlying the findings in their manuscript fully available?

Reviewer #1: Yes

Reviewer #2: No

4. Is the manuscript presented in an intelligible fashion and written in standard English?

Reviewer #1: Yes

Reviewer #2: Yes

5. Review Comments to the Author

Reviewer #1: In this paper, combining the DBO algorithm with the Otsu method, a new DSPO-OTSU image segmentation algorithm for rubber tree tangent and latex segmentation is proposed. In a comparative evaluation with six other meta-heuristic OTSU algorithms, namely SSA-OTSU, WOA-OTSU, WSO-OTSU, GGO-OTSU, AHA-OTSU, and CSA-OTSU, the results showed a 10% improvement in the structural similarity index.

Section 1.2 presents some challenges to this approach, but these challenges are not properly discussed in the results. In this paper, the DBO-Otsu method for rubber tree tangential and latex segmentation is introduced only by comparing the tangential results without giving a detailed analysis of latex segmentation. The paper also has the following problems：

m(k) in formula (0.6) represents m1(k) or m2(k)? Or some other meaning? Please give a specific explanation;

The specific meaning of the mathematical symbols in formula (8) , e.g., Jtmax, sigmoid and w need to be clarified.

Table 1 in section 4.4 is not found;

The introduction to Figure 1 is not presented;

The segmentation diagram of the traditional Otsu algorithm is shown in Figure 5, but the segmentation diagram of the DPO-OTSU method proposed in this paper is not given.

Figure 6 shows three different convergence graphs, but does not explain the difference between the three convergence graphs, only compares the difference between the DBO algorithm and other algorithms;

Figure 10-12 corresponds to the segmentation results of Figure 7-9 by different algorithms, but it is not introduced which original figure of Figure 7-9 corresponds to each column of results.

Does Figure 12 in line 9 on page 26 correspond to Figure 2 on page 36? Is it marked incorrectly?

Does row 7 in Figure 13 from the bottom of page 26 correspond to Figure 3 on page 37? Is it marked incorrectly?

The delivered figure is too blurred with lower resolution to distinguish. Please provide clearer figures and diagrams for readers. Meanwhile, a paper related to the rubber tree disease could be mentioned in the Introduction Section, which is mentioned as follows: Rubber Tree Crown Segmentation and Property Retrieval Using Ground-Based Mobile LiDAR after Natural Disturbances.

Reviewer #2: This paper presents a multi-level thresholding image segmentation model, DBO-Otsu, designed to address the need for precise classification and early diagnosis of Rubber Tree Top Wilt Disease. DBO-Otsu, based on Dung Beetle Optimization, outperforms standard Otsu methods, especially in high grayscale areas. The proposed DBO-Otsu is evaluated over three images from the rubber dataset as benchmark images. Comparative analysis with six other meta-heuristic Otsu algorithms shows a notable 10% enhancement in the Structural Similarity Index.

The paper highlights the utilization of image segmentation methods as a preprocessing step in the field of medical imaging, focusing on a timely and critical issue related to healthcare diagnostics.

The reviewer appreciates the effort invested in this manuscript. While the paper exhibits notable merits, there are significant improvements that should be considered before the paper becomes suitable for publication. Therefore, the reviewer recommends addressing these additional revisions.

Please see the attached file for detailed comments.

6. PLOS authors have the option to publish the peer review history of their article (what does this mean?). If published, this will include your full peer review and any attached files.

Reviewer #1: No

Reviewer #2: No

---

## [Author Response · Author response to Decision Letter 0]

29 Nov 2023

Zhenjing Xie

Hainan University

Haikou, 570228

China

2023.11.28

Khan Bahadar Khan, Ph.D

Academic Editor

PLOS ONE

Dear Editor Khan Bahadar Khan,

Thank you and the reviewers for the review and recommendations on our manuscript (Title: "Advancing Image Segmentation with DBO-Otsu: Addressing Rubber Tree Diseases through Enhanced Threshold Techniques," Manuscript ID: PONE-D-23-30775). We have made detailed revisions based on the reviewer's feedback. Below is our response to the reviewer's comments:

1) Major comments:

Introduction:

1. Comment: Begin the introduction with a brief overview of the Rubber Tree Top Wilt Disease and how cutting-edge AI techniques and computer vision play a pivotal role in identifying and understanding this disease. This will allow you to smoothly introduce image segmentation as a preprocessing step in medical imaging, setting the foundation for the subsequent discussion.

Response: Thank you for your detailed review and valuable suggestions regarding the introduction section of our paper. We recognize that the original introduction lacked the necessary cohesion, flow, and clarity of ideas. Following your guidance, we have thoroughly revised the introduction to better align with the theme and purpose of the paper.

In the revised introduction, we start with a brief overview of Rubber Tree Topping Panel Dryness (TPD), highlighting its significance to the global rubber industry. We further discuss the critical role of modern AI techniques and computer vision in identifying and understanding TPD. This approach allows us to smoothly introduce image segmentation as a preprocessing step in medical imaging, setting the foundation for the subsequent discussion.

Moreover, we emphasize the fundamental goal in image recognition, especially in dealing with the unique image characteristics of the cuts and latex, which is to achieve precise segmentation. High-quality image segmentation directly contributes to improved diagnostic accuracy, fostering objectivity and uniformity in evaluations. This approach aids in the automatic identification of TPD-affected rubber trees at different stages, assisting researchers in real-time monitoring and future latex yield prediction, and advancing the study of rubber tree diseases.

2. Comment: The authors mentioned that (traditional Otsu algorithms often face challenges in complex scenarios) Please clearly highlight the limitations of traditional Otsu algorithms in complex scenarios, to underscore the motivation for adopting metaheuristics as an alternative approach.

Response: Thank you very much for your thorough review and suggestions regarding our mention of the limitations of traditional Otsu algorithms in the introduction section of our paper. We realize that our original text did not sufficiently elucidate the limitations of traditional Otsu algorithms in complex scenarios. Therefore, following your guidance, we have enhanced and clarified this part in our revised manuscript.

In the revised introduction, we discuss in more detail the performance limitations of the Otsu algorithm in scenarios with highly variable background and target intensity, or significant noise disturbances. Particularly, its robustness is compromised in high-noise images affected by salt-and-pepper noise, leading to segmentation inaccuracies. We highlighted the novel improvements proposed to address these challenges.

We hope these revisions clearly underscore the motivation for adopting metaheuristic algorithms as an alternative approach and better explain the limitations of traditional Otsu algorithms.

3. Comment: Introduce thresholding-based techniques, emphasizing their popularity and relevance as similarity-based methods in the context of your research.

Response: Thank you very much for your suggestion regarding the introduction of thresholding-based techniques and emphasizing their popularity and relevance as similarity-based methods in the context of our research. We acknowledge that our original text did not adequately highlight this aspect. Therefore, following your guidance, we have made relevant enhancements and clarifications in our revised manuscript.

In the revised introduction, we have elaborated on threshold-based image segmentation techniques, emphasizing their simplicity and effectiveness, and their applicational value in our current research. Specifically, we mention that despite the superior performance of deep learning methods in image segmentation, their hardware requirements limit their application in wearable devices. Therefore, for Tapping Panel Dryness (TPD) recognition in rubber plantations, where environmental complexity and signal instability are issues, the lighter, more efficient threshold-based segmentation methods become a more appropriate choice.

We believe these additions and improvements not only provide a comprehensive presentation of the importance of threshold-based image segmentation techniques but also better contextualize them in our research.

4. Comment: Provide a more detailed explanation for the selection of Dung Beetle Optimization as the chosen metaheuristic approach and discuss why it was preferred over other available metaheuristics.

Response: Thank you very much for pointing out the need for a more detailed explanation of why we chose Dung Beetle Optimization (DBO) as the metaheuristic approach and its advantages over other available metaheuristic algorithms. We realize that our original text did not adequately articulate the specific reasons for this choice. Therefore, following your suggestion, we have made corresponding supplements and detailed explanations in our revised manuscript.

In the revised introduction, we elaborate on why we chose to combine the DBO algorithm with the Otsu method and its advantages in handling complex image segmentation challenges, such as rubber tree cuts and latex images. We emphasize that the DBO algorithm is inspired by the natural behaviors of dung beetles, such as their rolling and foraging strategies, which enhance the algorithm's performance in terms of search diversity, efficiency, and global convergence capabilities. Especially in multi-threshold image segmentation, DBO excels in convergence speed and solution accuracy compared to other metaheuristic algorithms.

We hope these additions clearly articulate why we chose the DBO algorithm as our optimization tool and better contextualize it in our research.

5. Comment: It is difficult to discern the unique contributions of the proposed model. It would be beneficial to highlight the contributions in bullet points. This would provide readers with a concise and structured overview of the specific advancements by the proposed model.

Response: Thank you for pointing out that the unique contributions of our proposed model were not clearly discernible and for suggesting that we highlight these contributions in bullet points. Indeed, our original text did not clearly list the specific advancements of the proposed model. Therefore, following your advice, we have highlighted these contributions in a more structured and concise manner in our revised manuscript.

In the revised introduction, we clearly outline the paper's four primary contributions in a bulleted list: (1) Introducing the DBO-Otsu algorithm to enhance the efficiency and accuracy of complex image segmentation; (2) Conducting a comprehensive evaluation of the DBO-Otsu algorithm using mainstream performance metrics; (3) Providing a theoretical analysis of the DBO algorithm, emphasizing its promising applications in image segmentation; (4) Demonstrating the algorithm's practical utility in real-world segmentation tasks, particularly in diagnosing rubber tree Tapping Panel Dryness. We also outline the structure of the paper.

In addition to the previously mentioned four primary contributions, we have further enriched the content of our paper. To more vividly demonstrate the advantages of our algorithm, the experimental steps, and the core strategies, we have included a comprehensive flowchart. This flowchart not only clearly delineates the operational process of the DBO-Otsu algorithm but also visually presents the various stages of the experiment and the detailed implementation of key strategies.

With this flowchart, we aim to provide readers with a more intuitive understanding, allowing them to grasp the overall picture and advantages of our method at a glance. We believe that this graphical representation not only enhances the readability of the paper but also helps deepen readers' understanding of our research.

Inadequate literature review

1. Comment: discussion of related works should be expanded with more recent metaheuristics-based image segmentation studies to highlight the limitations of the existing literature, situate the contribution, establish the research gap and highlight the novelty of the proposed approach.

Response: Thank you for suggesting that we expand the discussion of the integration of metaheuristic algorithms with the Otsu method in image segmentation to highlight the limitations of the existing literature, establish the research gap, and emphasize the novelty of our proposed approach. We realize that our original text did not sufficiently showcase these latest studies. Therefore, following your advice, we have included more recent related research in section 1.3 of our revised manuscript.

In these additions, we have detailed how the combination of metaheuristic algorithms with the Otsu method significantly enhances its capabilities. We listed a series of innovative studies such as DE-OTSU-GWO, FOA-OTSU, IGJO, AOA optimization of the Otsu method, LCGSA, and HCROA, which have made significant achievements in improving the accuracy, efficiency, and stability of image segmentation. We also emphasized how the DBO-Otsu algorithm, inspired by the natural behaviors of dung beetles, enhances search diversity, efficiency, and global convergence capabilities, outperforming other metaheuristic algorithms in multi-threshold image segmentation, particularly in convergence speed and solution accuracy.

We hope these additions provide readers with a comprehensive perspective on the current developments in the research field and clearly highlight the contributions and innovation of our proposed method.

2. Comment: Some recent related papers that could be included.

Response: Thank you for suggesting that we include recent related research papers in the introduction section of our manuscript. We realize that the inclusion of these references will enrich the research background of our paper and better showcase the current advancements in the field. Following your guidance, we have added several key recent papers in section 1.3 of our revised manuscript.

In these additions, we particularly highlight the latest advancements in swarm intelligence algorithms for multi-threshold segmentation, especially in their application to processing COVID-19 chest X-rays and CT scans. We have cited studies by Chen et al., Abualigah et al., Liu et al., Emam et al., and Chen et al., who have made significant contributions in enhancing initial convergence, global search efficiency, search efficiency, and segmentation precision, as well as in multi-threshold segmentation applications. These advancements not only advance the application of swarm intelligence in medical image processing but also offer potent tools for medical decision-making.

We hope these additions provide readers with a comprehensive perspective on the current developments in the research field. Thank you once again for your valuable suggestions, which have been instrumental in enhancing the depth and breadth of our work.

3. DBO-Otsu

1. Comment: Please expand the DBO-Otsu approach in Section 3, explaining how DBO is integrated with Otsu. Clarify the solution representation and fitness function. Consider using schematic views and visualizations for better clarity.

Response: We have expanded upon the DBO-Otsu method in Section 3, explaining how DBO is integrated with Otsu, and providing clear representations of the solution and the fitness function. We have used flowcharts for visualization to facilitate better elucidation.

2. Comment: Section 3.1 is a critical component of the manuscript which represent the main contribution. Hence, it is essential to highlight and justify the proposed enhancements.

Response: In Section 3.1, we have highlighted and substantiated the significance of the algorithmic enhancements proposed, where the improved formula is repeatedly applied in the subsequent pseudocode. The efficacy of these improvements will be further evidenced through experimental comparisons with the traditional OTSU algorithm, accentuating their importance. Moreover, in Section 3.2, we delve into the detailed integration principles of the DBO algorithm with the OTSU algorithm. Additionally, in Section 3.3, we discuss the judgment methods for specific levels and the refined strategies for segmenting latex, complemented by illustrative images to demonstrate their effective outcomes.

4. Experiments and analysis of results

1. Comment: It is not clear from the manuscript whether the parameter settings have been tuned specifically for your research or if they are adopted from recommended studies.

Response: 

In response to the issues you pointed out, we have made necessary revisions to enhance the clarity and technical depth of the paper. Specifically, we have provided more detailed explanations of the parameter settings used in the experiments.

In the original manuscript, we indeed did not explicitly indicate whether the parameter settings were specifically tailored for our research or adopted from recommended studies. To address this, we have explicitly stated in the revised manuscript that, apart from 'pop_num' (population size) and 'Max_iter' (maximum number of iterations), the other regional parameters were selected based on recommended research. We have adjusted 'pop_num' and 'Max_iter' to accommodate the complexity of the experiments while ensuring the reliability of our study under these specific conditions.

2. Comment: It is important to cite the utilized algorithms, specifying which versions of SSA, CSA, WOA, GWO, WSO, and AHA have been used in this research?

Response: To provide a more comprehensive comparative perspective, we have added a comparison with the CSA-Otsu method in the paper. All algorithms used for comparison, including SSA-Otsu, WOA-Otsu, WSO-Otsu, GWO-Otsu, AHA-Otsu, and CSA-Otsu, are accompanied by corresponding references, allowing readers to refer to the original research for the source and rationale of these parameter settings.

3. Comment: Please provide a brief description of the rubber dataset used as a benchmark in this study.

Response: We have provided a brief description of the rubber dataset used as the baseline, including its levels of Tapping Panel Dryness (TPD), and whether it was randomly selected or chosen under certain conditions.

4. Comment: While the selection of three images from the rubber dataset is a good starting point, it's important to note that this sample size may not be sufficient to comprehensively verify the efficiency of the proposed model. It would be beneficial, if possible, to expand the testing to include a more substantial number of images to ensure robust evaluation

Response: We have expanded the number of test samples to ensure comprehensive validation of the proposed model. This includes validation for both latex segmentation and tapping line segmentation. Furthermore, we have performed latex and tapping line segmentation on all images in the dataset, selected representative images for presentation in the paper, and provided the remaining images in the supplementary materials.

5. Comment: Please discuss the convergence curves of the proposed approach.

Response: We appreciate your valuable suggestions. Regarding the discussion of the convergence curves of the proposed DBO-Otsu method, we have provided a more detailed description and analysis in the revised manuscript.

In the original text before revision, we described the overall trend of the convergence curves and highlighted the rapid convergence speed of the DBO algorithm. However, upon your recommendation, we realized that the original description might not adequately showcase the detailed comparisons between various algorithms and the specific performance of the DBO-Otsu algorithm.

Therefore, in the revised manuscript, we not only present the convergence curves of different optimization algorithms but also provide in-depth analysis of these curves. Specifically, in Fig17, we explicitly demonstrate the convergence process of different algorithms and extensively discuss the performance of DBO-Otsu relative to other algorithms, such as SSA, under different experimental conditions. We found that, while in some experiments (e.g., 4-8, 4-19, and 4-20), other algorithms like SSA may perform better in the final results, DBO-Otsu exhibits strong performance in terms of convergence speed and accuracy, especially in the majority of test functions.

With these modifications, we aim to comprehensively showcase the performance of the DBO-Otsu algorithm and provide deeper insights into its performance under different conditions. We believe that these additions will help readers better understand the effectiveness and practicality of our approach.

6. Comment: Inadequate analysis and discussion of results: The results are not thoroughly analyzed. A more in-depth analysis of the results, including statistical analysis is needed.

Response: We conducted a more in-depth analysis of the results and applied the proposed method in practical scenarios. By statistically calculating the ratio of latex to tapping line pixels in the segmented images, we approximated the basis for assessing the level of Tapping Panel Dryness (TPD). We performed diagnosis on all the images in the dataset, and the diagnostic results are saved in Excel files within the folders corresponding to each level of TPD.

7. Comment: Statistical analysis is important to judge the significance of the findings.

Response: We have maximized our efforts to perform statistical analysis of the data for each experimental step and presented it in tabular form in the paper, with the excess data stored in the supplementary materials. In addition to increasing the statistical analysis of the previous experiments, we also referred to statistical analysis methods from other papers for use in the 'Application Evaluation' section, aiming to achieve clearer experimental results.

Conclusion:

Comment: The conclusion should be explored better and it needs to contemplate the eventual restrictions of the developed technique to address future works in this area.

Response: Thank you for your insightful comments and valuable suggestions regarding the conclusion section of our paper. We realize that the original manuscript did not adequately explore the potential limitations of the DBO-Otsu algorithm, nor did it thoroughly consider future directions in this field. Therefore, we have made significant improvements to the conclusion in our revised manuscript.

In the revised paper, we not only highlight the applicational value and efficiency advantages of the DBO-Otsu algorithm in the field of image segmentation but also specifically mention the challenges and limitations encountered in practical applications. For example, we point out that directly applying the DBO-Otsu algorithm might not suffice for all scenarios when dealing with images of rubber tree diseases at different stages. Hence, we adopted a more nuanced approach, conducting morphological analyses post-initial segmentation and tailoring strategies to images at various disease stages.

Moreover, we mention in the conclusion that while the DBO-Otsu algorithm is able to perform tasks accurately in most instances, there is a slight decline in accuracy in images of intermediate disease stages. We emphasize the necessity of optimizing the algorithm in future research, including considering the integration of edge detection algorithms to eliminate irrelevant areas on the trunk, to enhance segmentation accuracy.

We believe that these additions and improvements provide a comprehensive presentation of the advantages of the DBO-Otsu algorithm, honestly reflect its limitations, and offer clear directions for future research.

2) Minor notes:

1. Comment: Please verify that acronym deﬁnitions are provided upon their initial use in the manuscript.

Response: We ensured that definitions for all acronyms are provided upon their initial appearance.

2. Comment: Please enhance the quality of ﬁgures. it would be beneﬁcial to consider converting them into vector-based formats.

Response: We improved the quality of the figures and converted all the images in our paper into vector-based formats using Pace.

3. Comment: Please check some mistakes.

Response: We reviewed and corrected the errors in the manuscript.

We hope that these revisions meet the requirements of the reviewers and the editor, making our manuscript more suitable for publication. We look forward to your further guidance and feedback.

Sincerely,

Zhenjing Xie

Undergraduate Student

Hainan University

Haikou, 570228

China

20213005583@hainanu.edu.cn

---

## [Decision Letter · Decision Letter 1]

14 Dec 2023

PONE-D-23-30775R1Advancing Image Segmentation with DBO-Otsu: Addressing Rubber Tree Diseases through Enhanced Threshold TechniquesPLOS ONE

Dear Dr. Liu,

Thank you for submitting your manuscript to PLOS ONE. After careful consideration, we feel that it has merit but does not fully meet PLOS ONE’s publication criteria as it currently stands. Therefore, we invite you to submit a revised version of the manuscript that addresses the points raised during the review process.

We look forward to receiving your revised manuscript.

Kind regards,

Khan Bahadar Khan, Ph.D

Academic Editor

PLOS ONE

Journal Requirements:

Reviewers' comments:

Reviewer's Responses to Questions

**Comments to the Author**

1. If the authors have adequately addressed your comments raised in a previous round of review and you feel that this manuscript is now acceptable for publication, you may indicate that here to bypass the “Comments to the Author” section, enter your conflict of interest statement in the “Confidential to Editor” section, and submit your "Accept" recommendation.

Reviewer #2: (No Response)

Reviewer #3: All comments have been addressed

2. Is the manuscript technically sound, and do the data support the conclusions?

Reviewer #2: (No Response)

Reviewer #3: Yes

3. Has the statistical analysis been performed appropriately and rigorously? 

Reviewer #2: (No Response)

Reviewer #3: Yes

4. Have the authors made all data underlying the findings in their manuscript fully available?

Reviewer #2: (No Response)

Reviewer #3: Yes

5. Is the manuscript presented in an intelligible fashion and written in standard English?

Reviewer #2: (No Response)

Reviewer #3: Yes

6. Review Comments to the Author

Reviewer #2: Please see the attached file.........................................................................

Reviewer #3: The introduction would benefit from a more detailed description of the Rubber Tree Tapping Panel Dryness (TPD) disease and its implications within the rubber industry, thereby providing essential context for the study. A succinct presentation of the research gap and how this work aims to bridge it is also recommended for clarifying the research's significance.

7. PLOS authors have the option to publish the peer review history of their article (what does this mean?). If published, this will include your full peer review and any attached files.

Reviewer #2: No

Reviewer #3: No

---

## [Author Response · Author response to Decision Letter 1]

18 Dec 2023

Zhenjing Xie

Hainan University

Haikou, 570228

China

2023.11.28

Khan Bahadar Khan, Ph.D

Academic Editor

PLOS ONE

Dear Editor Khan Bahadar Khan,

Thank you and the reviewers for the review and recommendations on our manuscript (Title: "Advancing Image Segmentation with DBO-Otsu: Addressing Rubber Tree Diseases through Enhanced Threshold Techniques," Manuscript ID: PONE-D-23-30775). We have made detailed revisions based on the reviewer's feedback. Below is our response to the reviewer's comments:

1. Comment: The justification for selecting DBO over other MHs is still insufficient. It would be beneficial to emphasize the successful applications of DBO in the literature for addressing real-world problems. Additionally, please explicitly highlight the merits of DBO in comparison to other algorithms. Considering the problemdependent nature of metaheuristics and referring to the principles of the free lunch theorem would enhance the justification. Please include references.

Response: In response to your comment, Section 1.3 has been updated to include four recent publications featuring the application of the DBO optimization algorithm. These additions emphasize the suitability of the DBO algorithm for our research, particularly in terms of segmentation speed and accuracy, compared to other metaheuristic algorithms. Additionally, we have incorporated the 'No Free Lunch' theorem to acknowledge that no algorithm is universally perfect. This inclusion underscores the importance of selecting an algorithm that is most apt for the specific requirements of our study. Relevant references have been added to support these points.

2. Comment: It is difficult to discern the unique contributions of the proposed model. While the integration of a recent DBO with Otsu is good, it does not constitute a novel contribution. In this regard, I recommend highlighting the unique enhancement to the traditional Otsu (presented is section 3.1) explicitly in the list of contributions. Response: In light of your observation regarding the distinctiveness of our model's contributions, we have revised the final part of the Introduction to explicitly include the unique enhancements made to the traditional Otsu method in the list of contributions. Specifically, this update is reflected in the second point of the contributions section. This revision aims to clearly delineate our novel contribution, namely the integration of the recent DBO with Otsu, and its significance in advancing the methodology.

3. Comment: The authors state “Providing a theoretical analysis of the DBO algorithm”, but the presented work is primarily experimental. 

Response: In response to your comment on the theoretical analysis of the DBO algorithm, we have updated the contribution section by removing the reference to 'Providing a theoretical analysis of the DBO algorithm.' Acknowledging the primarily experimental nature of our work, we agree that mentioning a theoretical analysis could be misleading. This revision ensures that the contributions listed are in alignment with the experimental focus of our research.

4. Comment: Authors mentioned the use of an early convergence critera “During the iteration process, if the optimal threshold obtained is the same as the result of the first 5 iterations, the optimization process is considered to have converged and the iteration can be ended earlier, thus reducing the time consumption of the algorithm”. It is unclear whether this approach is applied to other compared algorithms. Additionally, the rationale behind choosing 5 iterations as the threshold for convergence needs clarification.

Response: Regarding the early convergence criteria mentioned in our manuscript, we acknowledge that the theoretical basis of such criteria is detailed in the paper 'Effective Probabilistic Stopping Rules for Randomized Metaheuristics: GRASP Implementations.' However, it is important to note that this is not the focal point of our current study. The specific approach of terminating the iteration process after five identical results in our study was based on empirical experience and has minimal impact on the experimental outcomes. Consequently, we have decided to remove this content from the original manuscript. Furthermore, we clarify that this early termination method has not been applied to the other algorithms compared in our research, ensuring consistency in methodology across all tested approaches.

5. Comment: Results in Table 2 does not demonstrate the superiority of the proposed approach in most cases. Authors mentioned that “In evaluating average PSNR, FSIM, and SSIM scores, DBO-OTSU consistently ranked at or near the top…” However, it is not clear if a specific ranking method was employed to support this claim. If so, kindly present the rank for each method. Utilizing a widely used ranking method like Friedman rank would be beneficial.

Response: To address your concern regarding the demonstration of our approach's superiority in Table 2, we have conducted a comprehensive statistical analysis. We calculated the average rankings for each algorithm in terms of PSNR, FSIM, and SSIM scores. These rankings have been incorporated into Section 4.3 in a tabular format, accompanied by a brief explanation. This addition provides a clearer and more intuitive comparison among the algorithms, especially in scenarios where the differences in scores are not significantly pronounced. The use of average rankings offers a more structured way to assess the relative performance of each method, allowing for a fair and transparent comparison.

6. Comment: How do the authors justify the claim that, despite achieving better fitness scores, certain algorithms may not perform better in terms of other evaluation metrics like PSNR, FSIM, and SSIM?

Response: To justify our claim regarding the discrepancy between achieving better fitness scores and the performance in other evaluation metrics like PSNR, FSIM, and SSIM, we clarify that maximizing fitness, which is the inter-class variance in the mid-to-high threshold range, primarily focuses on enhancing the contrast between the foreground and background in grayscale. While this can improve segmentation effectiveness, it does not necessarily translate to higher overall image quality. PSNR, FSIM, and SSIM evaluate the segmented multi-threshold image against the original image. The application of multi-threshold Otsu's method might lead to loss of certain details in the image, particularly in high-contrast areas and regions with complex textures like tree barks, potentially lowering PSNR. FSIM considers image phase congruency and gradient magnitude to assess visual quality. However, the algorithm alters local features, especially at edges and textures, affecting the FSIM score. SSIM measures the structural, luminance, and contrast similarity between two images. The segmentation changes brought by the algorithm can alter local structures, thereby impacting the SSIM score.

However, this should not be construed as diminishing the relevance of PSNR, FSIM, and SSIM scores. These metrics provide a comprehensive assessment of the segmented multi-threshold images from a conventional perspective, complementing and corroborating our specialized evaluation approach.

7. Comment: Statistical analysis is important to judge the significance of the findings. In this comment I mean that Utilizing tests such as the Wilcoxon rank sum test would be beneficial to determine whether the differences presented in Table 2, such as those in PSNR (e.g., 8.762745 and 8.783505), are statistically significant or not."

Response: To address the need for statistical analysis to ascertain the significance of our findings, we have augmented Section 4.3 with a table presenting the results of the Wilcoxon signed-rank test. This test compares the DBO algorithm with other algorithms, using a predefined significance level of 0.1. The p-values obtained from this analysis have been duly recorded. The results demonstrate that the DBO algorithm shows statistically significant improvements in the PSNR and SSIM metrics. However, for the FSIM metric, the improvements were not found to be statistically significant. This additional analysis provides a robust statistical basis for validating the superiority of the DBO algorithm in specific aspects of our study.

8. Comment: Please verify that acronym definitions are provided upon their initial use in the manuscript and consistently use the acronym in subsequent parts.

Response: We have meticulously reviewed the entire manuscript in response to your comment regarding the use of acronyms. We have corrected the instances where acronyms were incorrectly used or not defined upon their initial use. This ensures that all acronyms are now properly introduced and consistently applied throughout the manuscript, enhancing its readability and clarity.

9. Comment: Please double check some mistakes

Response: In response to your comment regarding the presence of errors, we have thoroughly re-examined the manuscript and have identified and corrected the mistakes at the relevant locations. We appreciate your attention to detail and are committed to ensuring the accuracy and quality of our work.

10. Comment: Please enhance the quality of figures. (still need improvements, for instance the DBO flowchart …..)

Response: In response to your feedback on the quality of the figures, specifically mentioning the DBO flowchart, we have undertaken a comprehensive redesign of this figure to enhance its clarity and overall presentation. We hope the revised figure now meets your expectations in terms of visual quality and effectiveness in conveying the intended information.

11. Comment: Please review the order and placement of figures to ensure that captions are appropriately associated with the corresponding figures throughout the manuscript.

Response: In response to your comment on ensuring appropriate alignment of figure captions with their corresponding figures, we have conducted a thorough review of the entire manuscript. This review focused on verifying the order and placement of all figures to guarantee that each caption accurately reflects and is correctly associated with its respective figure. We have made necessary adjustments to ensure this consistency throughout the document.

We hope that these revisions meet the requirements of the reviewers and the editor, making our manuscript more suitable for publication. We look forward to your further guidance and feedback.

Sincerely,

Zhenjing Xie

Undergraduate Student

Hainan University

Haikou, 570228

China

20213005583@hainanu.edu.cn

---

## [Decision Letter · Decision Letter 2]

3 Jan 2024

Advancing Image Segmentation with DBO-Otsu: Addressing Rubber Tree Diseases through Enhanced Threshold Techniques

PONE-D-23-30775R2

Dear Dr. Liu,

We’re pleased to inform you that your manuscript has been judged scientifically suitable for publication and will be formally accepted for publication once it meets all outstanding technical requirements.

Kind regards,

Khan Bahadar Khan, Ph.D

Academic Editor

PLOS ONE

Additional Editor Comments (optional):

Reviewers' comments:

Reviewer's Responses to Questions

**Comments to the Author**

1. If the authors have adequately addressed your comments raised in a previous round of review and you feel that this manuscript is now acceptable for publication, you may indicate that here to bypass the “Comments to the Author” section, enter your conflict of interest statement in the “Confidential to Editor” section, and submit your "Accept" recommendation.

Reviewer #2: All comments have been addressed

Reviewer #4: All comments have been addressed

2. Is the manuscript technically sound, and do the data support the conclusions?

Reviewer #2: Yes

Reviewer #4: Yes

3. Has the statistical analysis been performed appropriately and rigorously? 

Reviewer #2: Yes

Reviewer #4: Yes

4. Have the authors made all data underlying the findings in their manuscript fully available?

Reviewer #2: Yes

Reviewer #4: Yes

5. Is the manuscript presented in an intelligible fashion and written in standard English?

Reviewer #2: Yes

Reviewer #4: Yes

6. Review Comments to the Author

Reviewer #2: All comments have been adequately addressed

.................................................................................

Reviewer #4: The paper in its present second submission looks like fully ready to be considered for publication. All past comments have been fully addressed and more explanations have been added to the discussion. The results section has been apropriately widen as to include more information about the algorithm performance itself and its comparison to other algorithms is now clearer and more feasible.

7. PLOS authors have the option to publish the peer review history of their article (what does this mean?). If published, this will include your full peer review and any attached files.

Reviewer #2: No

Reviewer #4: **Yes: **Marco Perez-Cisneros

---

## [Editor Report · Acceptance letter]

25 Jan 2024

PONE-D-23-30775R2 

PLOS ONE

Dear Dr. Liu, 

I'm pleased to inform you that your manuscript has been deemed suitable for publication in PLOS ONE. Congratulations! Your manuscript is now being handed over to our production team.

Kind regards, 

on behalf of

Dr. Khan Bahadar Khan 

Academic Editor

PLOS ONE